# Enhanced basal melting in winter and spring: Seasonal ice–ocean interactions at the Ekström Ice Shelf, East Antarctica

Ole Zeising[1], Tore Hattermann[2], Lars Kaleschke[1], Sophie Berger[1,*], Olaf Boebel[1], Reinhard Drews[3], M. Reza Ershadi[3], Tanja Fromm[1], Frank Pattyn[4], Daniel Steinhage[1], and Olaf Eisen[1,5]

[1]Alfred-Wegener-Institut Helmholtz-Zentrum für Polar- und Meeresforschung, Bremerhaven, Germany
[2]Norwegian Polar Institute, Tromsø, Norway
[3]Department of Geosciences, Tübingen University, Tübingen, Germany
[4]Laboratoire de Glaciologie, Université Libre de Bruxelles, Brussels, Belgium
[5]Faculty of Geosciences, University of Bremen, Bremen, Germany
[*]Now at: European Commission

**Correspondence:** Ole Zeising (ole.zeising@awi.de)

**Abstract.** Basal melting of Antarctic ice shelves significantly contributes to ice sheet mass loss, with distinct regional disparities in melt rates driven by ocean properties. In Dronning Maud Land (DML), East Antarctica, cold water predominantly fills the ice shelf cavities, resulting in generally low annual melt rates. In this study, we present a four-year record of basal melt rates at the Ekström Ice Shelf, measured using an autonomous phase-sensitive radar (ApRES). Observations reveal a low mean annual melt rate of $0.44\,\mathrm{m\,a^{-1}}$, with a seasonal variability. Enhanced melting occurs in winter and spring, peaking at over $1\,\mathrm{m\,a^{-1}}$, while rates are decreased in summer and autumn. We hypothesise that the dense water formed during sea-ice formation erodes the water column stratification during late winter and spring, leading to an increase in the buoyancy of the ice shelf water plume. An idealised plume model supports this hypothesis, indicating that the plume velocity is the primary driver of seasonal basal melt rate variability, while changes in ambient water temperature plays a secondary role in the range of oceanographic conditions that are observed below the Ekström Ice Shelf. These findings offer new insights into the dynamics of ice-ocean interaction in East Antarctica, emphasising the need for further observations to refine our understanding of ocean variability within ice shelf cavities and improve assessments of ice-shelf mass balance.

## 1 Introduction

The Antarctic Ice Sheet is progressively losing mass, mainly due to the ocean-driven melting of the ice shelves in West Antarctica (Shepherd et al., 2018). Ice shelf thinning reduces the buttressing effect and causes the acceleration of outlet glaciers and ice streams (Dupont and Alley, 2005; Fürst et al., 2016; Reese et al., 2018). In contrast to West Antarctica, ice-shelf melting in East Antarctica is relatively low. Understanding the processes that contribute to basal melting is crucial to assessing the impact of a warming ocean on future ice shelf stability.

Jacobs et al. (1992) describes three modes in which ocean circulation affects basal melting. In mode 1 (cold-water cavity), the sea-ice formation in winter leads to dense shelf water with temperatures at the surface freezing point that sink to the seafloor. The dense water flows along the slope of the continental shelf which often deepens toward the grounding line. The dense water

drains beneath the ice shelf and delivers heat to the ice-shelf base since the temperature of the dense shelf water exceeds the pressure-melting point of the ice-shelf base. The buoyant plume of fresh meltwater mixes with the dense shelf water forming Ice Shelf Water (ISW) which rises along the basal slope of the ice shelf. Temperatures may fall below the freezing point if the pressure reduces sufficiently during the upward trajectory. The super-cooled ISW may freeze on the ice-shelf base as marine ice, or it flows out of the cavity, where it forms a semi-consolidated layer below the sea ice, called platelet ice (Hoppmann et al., 2020). In mode 2 (warm-water cavity), Warm Deep Water (WDW) rises above the continental shelf break and flows into the ice shelf cavity, where it can cause high basal melt rates. In mode 3, warm and fresh (and hence less dense) Antarctic Surface Water (AASW) in summer is transported beneath the ice shelf front, leading to high basal melt rates (Hattermann et al., 2012; Lindbäck et al., 2019; Stewart et al., 2019).

Observing ice–ocean interactions is challenging due to the inaccessibility of the ice shelf–ocean interface. Oceanographic moorings beneath or in front of ice shelves allow the observation of ocean properties that can reveal water masses and currents and their temporal variability. Due to the logistical effort required for the deployment and the risk of destruction by passing icebergs in case of deployment in front of the shelves, these moorings are rare. However, ice shelf basal melting can be estimated over large areas by satellite remote sensing (e.g. Rignot et al., 2013; Adusumilli et al., 2020; Davison et al., 2023). The analysis of the time-averaged melt rate pattern allows conclusions to be drawn about whether there is rather cold or warm water in the sub-ice-shelf cavity, which could be linked to one of the three modes. High melt rates near the grounding line and near the ice front, accompanied by low melt rates or refreezing in between, indicate the presence of a cold-water cavity characterized by mode 1 and mode 3 melting. Such a pattern is found, among others, on the largest Antarctic ice shelves: Ross, Ronne, Filchner and Amery (Adusumilli et al., 2020). When high melt rates extend across the entire ice shelf, this indicates warm-water cavities with mode 2 melting as found in the Amundsen and Bellinghausen seas in West Antarctica (Adusumilli et al., 2020).

Numerous small ice shelves exists in Dronning Maud Land (DML) in East Antarctica, whose drainage basins translate into a sea-level equivalent of over $3\,\mathrm{m}$ (Rignot et al., 2019). The driver of mode-1 melting is Eastern Shelf Water (ESW) with temperatures close to the surface freezing point (Vernet et al., 2019). However, ESW is less dense than the WDW that is observed to enter through topographic depressions at the continental shelf break (Lauber et al., 2023). It provides additional heat for melting, while the solar heated and relatively fresh AASW may intrude deep below the ice shelf (Hattermann et al., 2014; Lauber et al., 2024) and affect the cavity circulation on seasonal timescales (mode 3).

Basal melt rates from satellite remote sensing revealed in general low melt rates of $0.8\pm0.3\,\mathrm{m\,a^{-1}}$ on average but higher melt rates exceeding $5\,\mathrm{m\,a^{-1}}$ in the freely floating areas near the deep grounding lines of the Jelbart, Fimbul and Roi Baudouin ice shelves, where the local melting point temperature is reduced due to the higher pressure (Adusumilli et al., 2020). The outflow of potentially supercooled ISW (Nøst et al., 2011), as well as the accretion of significant amounts of platelet ice beneath coastal land fast ice (Arndt et al., 2020) have also been observed in the Atka Bay. However, for most of the DML ice shelves, a detailed understanding of the year-round ice shelf cavity circulation, and the interaction of the different modes of melting is still lacking.

For a more detailed investigation of the ice–ocean interaction, especially with a higher temporal resolution, other methods must therefore be used. From oceanic conductivity, temperature and depth (CTD) observations, Hattermann et al. (2012)

found a cold-water cavity at Fimbul Ice Shelf with temporal inflow of modified WDW (mWDW, mode 2) and fresh surface water in summer and autumn (mode 3) that increase basal melt rates. A complementary method for studying basal melting is phase-sensitive radio-echo sounding. This technique enables the observation of ice thickness changes over a specified period,
allowing the determination of basal melt rates based on the acquired data. Three studies in Dronning Maud Land used such an autonomous ground-based radar system that allows the determination of the basal melt rate with high temporal resolution. At the Nivl Ice Shelf, Lindbäck et al. (2019) found a seasonal variation in the melt rate $4\,\mathrm{km}$ from the ice shelf front with highest melt rates of up to $5.6\,\mathrm{m\,a^{-1}}$ in summer ($0.8\,\mathrm{m\,a^{-1}}$ annual average), indicating that warm surface water was pushed in the ice shelf cavity (mode 3). At $35\,\mathrm{km}$ from the ice shelf front, no seasonality was observed, although the highest melt rate occurred in
winter. Lindbäck et al. (2025) revealed a minor seasonal variation approximately $70\,\mathrm{km}$ from the ice shelf's front of the Fimbul Ice Shelf. Melt rates experienced a slight increase of approximately $0.3\,\mathrm{m\,a^{-1}}$ (33%) between spring and autumn compared to the winter months. This increase was attributed to elevated ocean velocities and water temperature near the base, indicating an interplay of AASW inflows (mode 3) and ISW outflows (mode 1) (Lauber et al., 2024). Sun et al. (2019) observed the basal melt rate near the grounding line of Roi Baudouin Ice Shelf and found nearly no melting in winter and highest melt rates of up
to $10\,\mathrm{m\,a^{-1}}$ in summer. No oceanic measurements were available to determine whether WDW could reach this location, but the moderate melt rates indicate primarily a mode-1 driven melting near the grounding line, while the analysis of Sun et al. (2019) hypothesised that the melt rate variability was caused by seasonally enhanced propagation of tidal oscillations into the ice shelf cavity, which increase the turbulent heat transfer near the ice base. Accordingly, the three ice shelves all show increased melt rates in summer, partly on the ice shelf front (Fimbul and Nivl), but also near the grounding line (Roi Baudouin), albeit due to
different processes. This raises the question of temporal melt rate variability and the responsible processes at other ice shelves in DML.

The Ekström Ice Shelf is located in the western Dronning Maud Land, East Antarctica, and covers an area of $\sim 6800\,\mathrm{km}^2$ (Neckel et al., 2012). The two ice rises Søråsen and Halvfarryggen confine the Ekström Ice Shelf and partly separate it from the two smaller ice shelves Quar and Atka (Fig. 1). The ice thickness varies from $1000\,\mathrm{m}$ at the grounding line to $250\,\mathrm{m}$ at
80 the calving front, where the ice velocity is roughly $220\,\mathrm{m\,a^{-1}}$ (Mouginot et al., 2019a, b; Morlighem et al., 2020; Morlighem, 2022). The bathymetry below the ice shelf consists of an inland-sloping trough that reaches a maximum depth of $1100\,\mathrm{m}$ below sea level near the grounding line (Eisermann et al., 2020; Smith et al., 2020). With a depth of $320\,\mathrm{m}$, the continental shelf break is sufficiently shallow to prevent the inflow of WDW into the cavity below Ekström Ice Shelf (Eisermann et al., 2020). The water column thickness in the cavity below the Ekström Ice Shelf ranges from $220\,\mathrm{m}$ at the front to $> 400\,\mathrm{m}$ for larger parts in
the centre of the trough (Eisermann et al., 2020; Smith et al., 2020). Observations from sub-ice-shelf CTD profiles taken within the Sub-EIS-Obs project $30\,\mathrm{km}$ from the ice front showed that the cavity below Ekström Ice Shelf is filled with relatively cold ESW (in situ temperatures of $-1.9\,^\circ\mathrm{C}$, practical salinity of $34.4$), while a buoyant plume of ISW (colder, less saline) is present near the ice-shelf base (Smith et al., 2020). The WDW potentially provides ocean heat for ice shelf melting. However, since the WDW is suppressed by the Antarctic Slope Front to depths around $500\,\mathrm{m}$ (Hattermann, 2018), which is generally deeper than
the continental shelf break in this region (Smith et al., 2020), WDW has not been observed inside the cavity of the Ekström Ice Shelf. Observations along the front of the Dronning Maud Land ice shelves (Nøst et al., 2011) and below the neighbouring

Fimbul Ice Shelf (Hattermann et al., 2012; Lauber et al., 2024) indicate that remnants of fresher, solar-heated AASW may enter several hundred meters below the ice draft in this region.

Estimated basal melt rates of the Ekström Ice Shelf from satellite remote sensing methods are relatively low. Adusumilli et al. (2020) found an average basal melt rate of $1.0 \pm 1.2\,\mathrm{m\,a^{-1}}$ (period 2010–2018) over the entire ice shelf and a maximum melt rate exceeding $3\,\mathrm{m\,a^{-1}}$ near the grounding line as well as near the German station Neumayer III (Fig. 1c). For the period between 1996 and 2006, Neckel et al. (2012) found melt rates of $< 1.1\,\mathrm{m\,a^{-1}}$ with highest melt rates near the grounding line.

While previous studies have provided insights into the basal melt rates of ice shelves in Dronning Maud Land, a detailed understanding of their temporal variability and the governing oceanic processes remains limited. In this study, we address this gap by analysing a four-year time series of basal melt rates obtained from an autonomous phase-sensitive radar located $25\,\mathrm{km}$ upstream of the ice shelf front. To further investigate the oceanic drivers of melt rate variations, we incorporate sea-ice growth rates, ocean temperature and salinity measurements, and an idealized plume model to assess the processes controlling basal melting at the Ekström Ice Shelf.

## 2 Data and methods

### 2.1 Basal melt rates

To measure basal melt rates, we operated an autonomous phase-sensitive radio-echo sounder (ApRES, Brennan et al., 2014; Nicholls et al., 2015) on the Ekström Ice Shelf $25\,\mathrm{km}$ upstream of the ice shelf front (Fig. 1b; location at $70.82\,°\mathrm{S}$, $8.73\,°\mathrm{W}$ on 02 April 2020 termed MIMO-EIS8). An ApRES is a ground-penetrating frequency-modulated continuous-wave (FMCW) radar that transmits chirps throughout $1\,\mathrm{s}$ while it increases the frequency of the electromagnetic wave from $200$ to $400\,\mathrm{MHz}$ (Nicholls et al., 2015).

The ApRES and two skeleton slot antennas were placed below the surface for protection from wind and snowdrifts. It performed hourly measurements consisting of 20 chirps between April 2020 and November 2023. Short gaps of two to three days exist in the recorded time series (09–10 January 2021, 25–26 November 2021, 31 December 2022 – 02 January 2023), at which the device was dismounted and used for other measurements, and a longer gap from 1 April to 16 May 2021.

The signal processing we applied to the raw data follows the previous work by Brennan et al. (2014), Nicholls et al. (2015) and Stewart et al. (2019). First, we rejected the first chirp of each measurement and stacked the remaining chirps to get a better signal-to-noise ratio. Next, to obtain an amplitude and phase profile as a function of two-way travel time from the stacked signal, we applied a Fourier transformation. Finally, we obtained a range profile by converting the two-way travel time into a range using a propagation velocity of $168\,914\,\mathrm{km\,s^{-1}}$ according to the ordinary relative permittivity of 3.15 (Fujita et al., 2000). The sample spacing due to zero-padding with a pad-factor of 8 is $0.053\,\mathrm{m}$, while the resolution of the phase is about $1°$, corresponding to $1\,\mathrm{mm}$ in range.

The determination of the basal melt rate from an ApRES time series is based on the analysis of the ice thickness change and the decomposition of the processes contributing to it (Nicholls et al., 2015). The change in ice thickness $\Delta H$ within the time

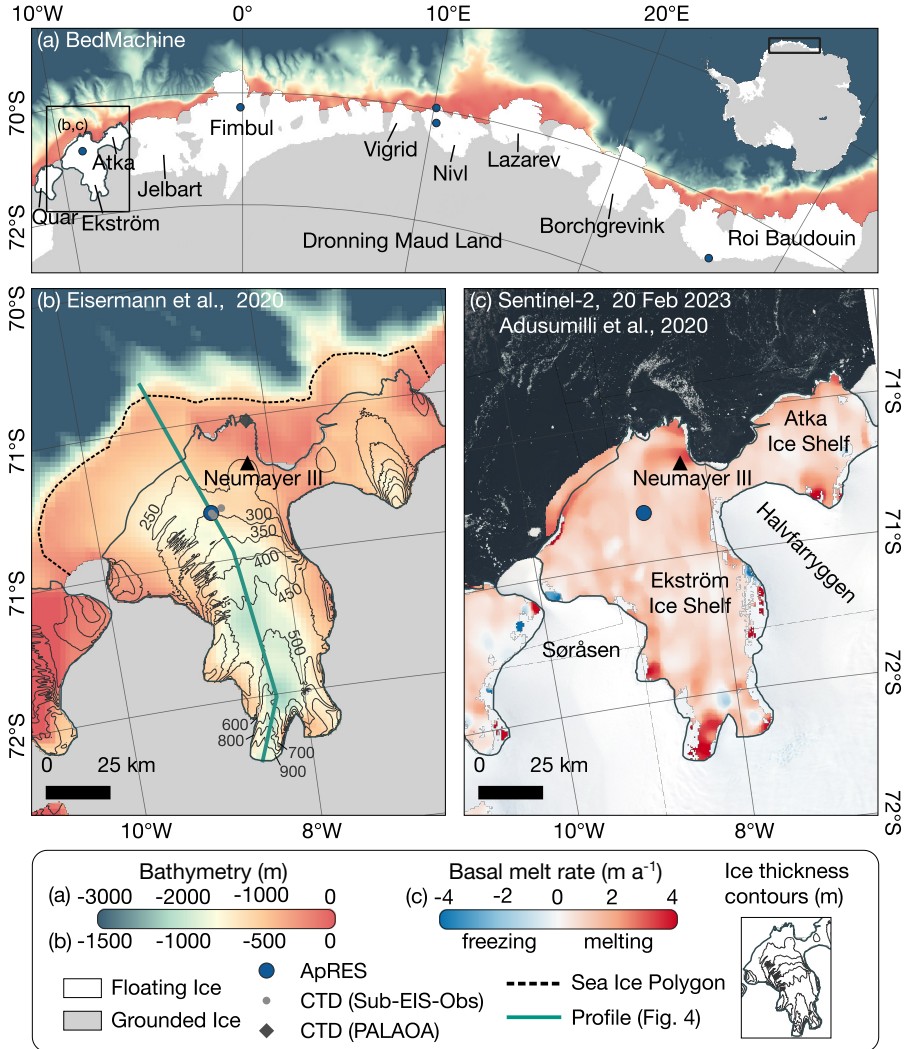

**Figure 1.** (a) Overview of Dronning Maud Land's ice shelves with the bathymetry in front of ice shelves from BedMachine V3 (Morlighem et al., 2020; Morlighem, 2022). The blue dots show the location of the ApRES measurements at Ekström Ice Shelf (this study), Fimbul Ice Shelf (Lindbäck et al., 2025), Nivl Ice Shelf (Lindbäck et al., 2019) and Roi Baudouin Ice Shelf (Sun et al., 2019). (b) Bathymetry below and in front of Ekström Ice Shelf from Eisermann et al. (2020) with ice thickness contours from BedMachine. The black triangle marks the location of the German station Neumayer III. The green line represents the location of the profile shown in Fig. 4. The dashed line marks the polygon within which the sea-ice growth has been estimated. The grey dots show the CTD locations from the Sub-EIS-Obs project (Smith et al., 2020) and the rhombus shows the PALAOA site. (c) Copernicus Sentinel-2 image from 20 February 2023, retrieved from the Copernicus SciHub on 31 July 2023, overlain by satellite-derived basal melt rates from Adusumilli et al. (2020) for the period 2010 to 2018.

interval $\Delta t$ is the sum of the thickness changes caused by firn compaction $\Delta H_f$, vertical strain $\Delta H_\varepsilon$ and basal melting $\Delta H_b$:

$$\frac{\Delta H}{\Delta t} = \frac{\Delta H_f}{\Delta t} + \frac{\Delta H_\varepsilon}{\Delta t} + \frac{\Delta H_b}{\Delta t}. \tag{1}$$

Solving Equation (1) for the melt rate $a_b$ gives:

$$a_b = -\frac{\Delta H_b}{\Delta t} = -\left( \frac{\Delta H}{\Delta t} - \frac{\Delta H_f}{\Delta t} - \frac{\Delta H_\varepsilon}{\Delta t} \right), \tag{2}$$

where positive $a_b$ represents basal melting.

For the melt-rate analysis, we follow the method described by Zeising and Humbert (2021) and Humbert et al. (2022), which are based on the previous work by Nicholls et al. (2015) and Stewart et al. (2019). The ApRES time series allows the determination of $\Delta H$, $\Delta H_f$, and $\Delta H_\varepsilon$ with millimetre precision. To this end, we divided the range profile into $6\,\mathrm{m}$ long segments with $3\,\mathrm{m}$ overlap below a depth of $10\,\mathrm{m}$. Around the basal reflection, identified by a strong increase in amplitude, we used a larger segment of $10\,\mathrm{m}$ size ($-9\,\mathrm{m}$ to $+1\,\mathrm{m}$ with respect to the basal reflection). For the basal segment, we derived displacements from complex-valued cross-correlation of the phase of all pairwise time-consecutive measurements. For each internal segment, we obtained the displacement time-series from a cross-correlation of the complex signal of the first measurement ($t_1$) with each repeated measurement ($t_i$). We used the lag of the correlation coefficient with the highest amplitude to find the correct phase-shift minimum, the sum of which gives the vertical displacement of this segment since the time of the first measurement. During the three interruptions in the ApRES time series in January 2021, November 2021, and December 2022/January 2023, the ApRES and the antennas were locally repositioned, a few metres laterally away from the previous location and placed again close to the surface to avoid burial over time. Therefore, we redefined $t_1$ as the first measurement after these interruptions.

The change in ice thickness and the processes contributing to it can now be determined based on the vertical displacements. The cumulative change in ice thickness results from the time series of the displacement of the basal segment. The contributions of firn compaction and vertical strain must be calculated based on the vertical displacement $u_z$ of various englacial segments. The vertical strain $\varepsilon_{zz}$ is defined as the vertical derivative of the vertical displacement $u_z$:

$$\varepsilon_{zz} = \frac{\partial u_z}{\partial z}. \tag{3}$$

We calculated the derivative of the vertical displacements based on a linear regression analysis assuming constant strain over depth as expected for the shallow-shelf flow regime of ice shelves. Here, we considered only those segments below the range of $100\,\mathrm{m}$ and $30\,\mathrm{m}$ above the basal return to avoid firn compaction and the basal return to influence the strain rate analysis. The linear regression analysis gives the cumulative displacement

$$u_z(z) = \varepsilon_{zz} z + \Delta H_f, \tag{4}$$

where the mean firn compaction $\Delta H_f$ is the intercept at the surface. The strain contribution to the change in ice thickness $\Delta H_\varepsilon$ is then the integral over the ice thickness $H$:

$$\Delta H_\varepsilon = \int_0^H \varepsilon_{zz}\, \mathrm{d}z. \tag{5}$$

Appendix A shows the analysis of the basal melt rate for one example from 2020 and the cumulative time series. We obtained the time series of cumulative melt by subtracting the contributions of firn compaction and strain thinning from the cumulative

ice thickness change. To determine the melt rate, we calculated the gradient in a $7\,\mathrm{d}$ moving window, which gives the $7\,\mathrm{d}$ average melt rate. By applying the $7\,\mathrm{d}$ moving window, variations with periods shorter than $\sim 16\,\mathrm{d}$ are attenuated.

To represent the variability of the melt rate on sub-weekly time scales, we additionally calculated the melt rate using the gradient in a $1\,\mathrm{d}$ moving window and then calculated the $7\,\mathrm{d}$ standard deviation. The uncertainty of the vertical displacements is based on the standard error of the radar phase which is in the order of sub-millimetres. The largest uncertainty originates from the strain analysis and its extrapolation to the ice base. The ApRES measurements at Ekström Ice Shelf allow the determination of the depth profile of displacements from near the surface to about $90\%$ of the ice thickness. The only assumption we made here is that the strain in the lowest $10\%$ of the ice thickness is the same as above. Since the measurement location was on a freely floating part of an ice shelf, the vertical displacements are sufficiently represented by a constant strain. Thus, it is accurate to assume that the strain remains constant in the remaining $10\%$ of the ice thickness. On average the uncertainty of the basal melt rate amounts to $\sim 0.02\,\mathrm{m\,a^{-1}}$.

## 2.2 Sea-ice growth and concentration

To investigate the relation between sea-ice formation and an increase in melt rate, we determined sea-ice growth and concentration over the continental shelf in front of the Ekström and Atka ice shelves (Fig. 1b). Sea-ice growth was estimated using the simple model of Pease (1987), which relates the sea-ice growth to the heat loss at the ice-free ocean surface in a polynya. We used surface wind speed and air temperature from JRA-55 reanalysis (Kobayashi et al., 2015) to calculate heat loss over the ice-free ocean. The sea-ice concentration and open water area were derived from AMSR-2 passive microwave data twice daily on a $3.125\,\mathrm{km}$ grid (Beitsch et al., 2014). Time series of sea-ice growth and concentration were derived within a fixed polygon area (Fig. 1b) for the same period as the ApRES measurements.

This study employs a heat budget method similar to previous approaches (e.g. Ohshima et al., 2016; Tamura et al., 2016) but with differences in methodology and satellite data. While this method likely overestimates sea-ice production due to simplifications such as neglecting ocean heat flux and solar radiation, it effectively captures temporal variability and seasonality. Comparisons with other resolutions and reanalysis forcing reveal consistent patterns in variability but differences in magnitude by up to a factor of two.

## 2.3 Ocean temperature, salinity and density

No observations of the oceanographic properties near the ApRES location have been made, except for a snapshot from the Sub-EIS-Obs ice shelf cavity CTD profiles taken in January 2019 (Smith et al., 2020). To assess the seasonal evolution of water masses within the cavity, we analysed hydrographic data from a CTD mooring that was installed in December 2005 as part of the „PerenniAL Acoustic Observatory in the Antarctic Ocean" (PALAOA; Boebel et al., 2006) near the front of the Ekström Ice Shelf (Fig. 1b). The CTD was deployed by means of hot-water drilling through the approximately $100\,\mathrm{m}$ thick ice shelf. At the time of the deployment, the ice shelf cavity had a thickness of approximately $160\,\mathrm{m}$ (sea-floor to ice shelf base), and the CTD was initially located $70\,\mathrm{m}$ beneath the ice shelf base. Due to melting and advancement of the ice shelf, these initial measurements underwent slight variations over time. PALAOA yielded an almost eight-year long record of pressure,

temperature and salinity from January 2006 to May 2014. The data record is interrupted for most of 2006, due to technical
problems of the CTD's cable. It was repaired in January 2007, after which records were uninterrupted until the end of the time
series in May 2014. The frozen-in CTD cable prevented the retrieval of the CTD and thus its post-deployment calibration. The
CTD was operated in stand-alone mode, with the sensor automatically activated every 30 minutes (Klinck, 2008). After 09
January 2012, the sampling rate was increased to once every 10 minutes.

In-situ measurements from the Sub-EIS-Obs and PALAOA CTDs were converted to conservative temperature and absolute
salinity (referred to temperature and salinity in the following for simplicity) using the TEOS-10 Gibbs thermodynamic potential
for seawater (Feistel, 2008), which was also used to derive potential density from the observations. To compare the temperature,
salinity, and density time series with the later observed melt rates, sea-ice growth and concentration, we calculated seasonal
composites of all products based on the multi-year data sets. For that purpose, we sorted the data according to the day of the
year and computed the median, and $25\%$ and $75\%$ percentiles within weekly bins throughout the year.

The PALAOA CTD is located approximately $40\,\mathrm{km}$ north-east of the ApRES location. Tidally modulated water flow veloc-
ities of $0.4$–$0.5\,\mathrm{m\,s^{-1}}$ were inferred at this site (Ivanciu, 2014), with evidence of intense mixing below the northeastern part of
the ice shelf (Smith et al., 2020). Tidal model simulations (Padman et al., 2008) that align with observations at the Ekström
Ice Shelf (Fromm et al., 2023) predict spring tide velocities of $0.2$–$0.3\,\mathrm{m\,s^{-1}}$ in the vicinity of the ApRES location. Given
that the tidal energy is concentrated at the diurnal and semi-diurnal frequencies, this indicates tidal excursions in the order of
$10\,\mathrm{km\,d^{-1}}$, suggesting that water masses from the PALAOA site may influence the ApRES location at time scales of days.

On horizontal scales that exceed a few tidal excursions, the diffusive nature of such tidal transports is expected to become less
important and, the coherent background flow becomes more relevant. Inside ice shelf cavities, buoyancy forces resulting from
water mass transformation due to basal melting may induce a cavity-wide overturning circulation (Determann and Gerdes,
1994). With typical flow velocities ranging from $0.1$ to $0.2\,\mathrm{m\,s^{-1}}$, the advection time scale between the ice front and the
approximately $140\,\mathrm{km}$ distant grounding line of the Ekström Ice Shelf is in the order of $8$ to $16\,\mathrm{d}$.

## 2.4 Plume model

To investigate the impact of varying ocean properties on basal melt rates beneath the Ekström Ice Shelf, we employed the ide-
alised ice-shelf plume model of Jenkins (1991), which has previously been used to describe larger-scale circulation beneath ice
shelves. Assuming that circulation and mixing beneath the ice shelf are primarily driven by thermohaline processes, the model
describes a buoyant meltwater-laden plume that rises along the sloping ice base. The plume expands through the entrainment
of ambient ocean waters, and the heat brought into the plume, as a result, it drives melting at the ice–ocean interface. Concep-
tually, the plume gains buoyancy because the meltwater input reduces its density compared to the salinity stratified ambient
ocean, while melt rates are parameterised as a function of the plume velocity and the difference between the plume temperature
and local melting point based on a shear-driven turbulent heat flux formulation. Omitting Coriolis' effect, the system is param-
eterised along a one-dimensional flowline, described by four ordinary differential equations with prognostic variables for the
plume thickness $D$, speed $U$, temperature $T$, and salinity $S$ with details of the formulation being given in Jenkins (1991). The

model is steady in time, uniform in the across-flow direction, and depth-integrated, leaving the along-track distance $X$ as the only independent variable (Jenkins, 2011).

The Ekström Ice Shelf comprises a confined ice geometry along a quasi-one-dimensional flowline. The plume model approximates the averaged cavity circulation across the ice flow to provide an initial assessment of the dynamics of buoyant ISW that rises along the upward-sloping ice base. However, the three-dimensional circulation beneath the ice shelf is likely influenced by rotational effects. Consequently, the model results are only applicable to regions where the flow is constrained by topography (Jenkins, 1991). Further limitations of this approach will be addressed in the discussion. The plume dynamics are largely governed by the slope of the ice base and the density difference between the plume and the ambient ocean properties. The former is prescribed based on BedMachine (Morlighem et al., 2020; Morlighem, 2022), and a seismic profile running from the grounding line to the ice front of the Ekström Ice Shelf (Smith et al., 2020).

To investigate the impact of varying water mass properties on the cavity circulation and melt rates, we performed simulations employing different vertical profiles of ambient temperature and salinity. During late winter and spring, convection after sea-ice formation generates a well-mixed body of Winter Water (WW) that occupies the continental shelf (Nøst et al., 2011). In late summer and autumn, stratified AASW enters the cavity, increasing the temperature and reducing the salinity (and consequently the density) in the upper portion of the water column below the ice shelf (Hattermann et al., 2012). Two profiles representing the seasonal extremes were constructed using a constant water mass end-member with a bottom temperature of $-1.85\,°C$, salinity of $34.4\,\mathrm{g\,kg^{-1}}$, which resembles the deep part of the Sub-EIS-Obs CTD profiles that is assumed to be approximately representative for the year-round conditions in the lower part of the cavity. This end member was linearly interpolated to the observed water masses at $150\,\mathrm{m}$ derived from the PALAOA CTD time series (and with temperature and salinity kept constant above to the surface), representing a simplified late winter/autumn (Sept/Oct) WW extreme and a late summer/spring (Feb/Mar) AASW extreme. These values align with the observed seasonality in the open ocean in front of the DML ice shelves (Nøst et al., 2011) and below the neighbouring Fimbul Ice Shelf (Hattermann et al., 2012).

## 3 Results

### 3.1 Observed basal melt rates

The ApRES time series from 2020 to 2023 reveals a low melt rate, averaging $0.44 \pm 0.30\,\mathrm{m\,a^{-1}}$ (mean $\pm$ standard deviation) with a seasonal melt pattern (Tab. 1, Figs. 2, 3a). Basal melt rates are reduced to a minimum of $0.25 \pm 0.16\,\mathrm{m\,a^{-1}}$ in summer (December, January, February; DJF; 2020–2023 average), they increase slightly to $\sim 0.4\,\mathrm{m\,a^{-1}}$ in autumn (March, April, May; MAM) and during winter (June, July, August; JJA), and reach their maximum in spring (September, October, November; SON) with an average of $0.69 \pm 0.30\,\mathrm{m\,a^{-1}}$ (Tab. 1, Fig. 2).

A variability on monthly time scales is most pronounced when the highest melt rates occur in winter and spring. During this time of the year, the difference between the minimum and maximum melt rate within a 28-d period exceeds $1\,\mathrm{m\,a^{-1}}$. In summer and autumn, the difference within the same time window is less than $0.3\,\mathrm{m\,a^{-1}}$. The largest melt events that are observed from July/August until October/November, occurred with intervals ranging from two to four weeks, with an average duration of $21\,\mathrm{d}$.

At the onset of these events, the 7-day averaged melt rate increased to $1\,\mathrm{m\,a^{-1}}$ or more within several days. The strongest melt event occurred in September 2020, when the 7-day averaged melt rate increased from $0.21$ to $1.83\,\mathrm{m\,a^{-1}}$. After two weeks, the melt rate decreased again to $0.54\,\mathrm{m\,a^{-1}}$, before the next melt event occurred. The seasonality was strongest in 2020 with on average 5 times higher melt rates in spring than in summer 2020/21 (Tab. 1). In 2021 and 2022, we see a weaker seasonality with only 2 times higher melt rates in spring compared to summer. The trend of weaker seasonality from 2020 and 2022 is caused by lower melt rates in spring that decreased from $1.08 \pm 0.31\,\mathrm{m\,a^{-1}}$ in 2020 to $0.54 \pm 0.34\,\mathrm{m\,a^{-1}}$ in 2022.

**Table 1.** Mean value and standard deviation of basal melt rate separated for austral autumn (MAM), winter (JJA), spring (SON), and summer (DJF). The annual mean represents the March–February average value.

**Basal melt rate** $(\mathrm{m\,a^{-1}})$

| Year | 2020/21[*] | 2021/22 | 2022/23 | 2023/24[**] | Average |
|---|---|---|---|---|---|
| Autumn | $0.27 \pm 0.09$ | $0.33 \pm 0.10$ | $0.48 \pm 0.14$ | $0.40 \pm 0.16$ | $0.37 \pm 0.12$ |
| Winter | $0.31 \pm 0.20$ | $0.52 \pm 0.38$ | $0.44 \pm 0.30$ | $0.37 \pm 0.22$ | $0.41 \pm 0.28$ |
| Spring | $1.08 \pm 0.31$ | $0.60 \pm 0.29$ | $0.54 \pm 0.34$ | $0.54 \pm 0.28$ | $0.69 \pm 0.30$ |
| Summer | $0.24 \pm 0.14$ | $0.26 \pm 0.19$ | $0.25 \pm 0.16$ | – | $0.25 \pm 0.16$ |
| Annual | $0.47 \pm 0.42$[*] | $0.43 \pm 0.28$ | $0.43 \pm 0.22$ | $0.44 \pm 0.22$[**] | $0.44 \pm 0.30$ |

[*]In 2020/21, the measurements cover only the months of April 2020 to February 2021.

[**]In 2023/24, the measurements cover only the months of March 2023 to October 2023.

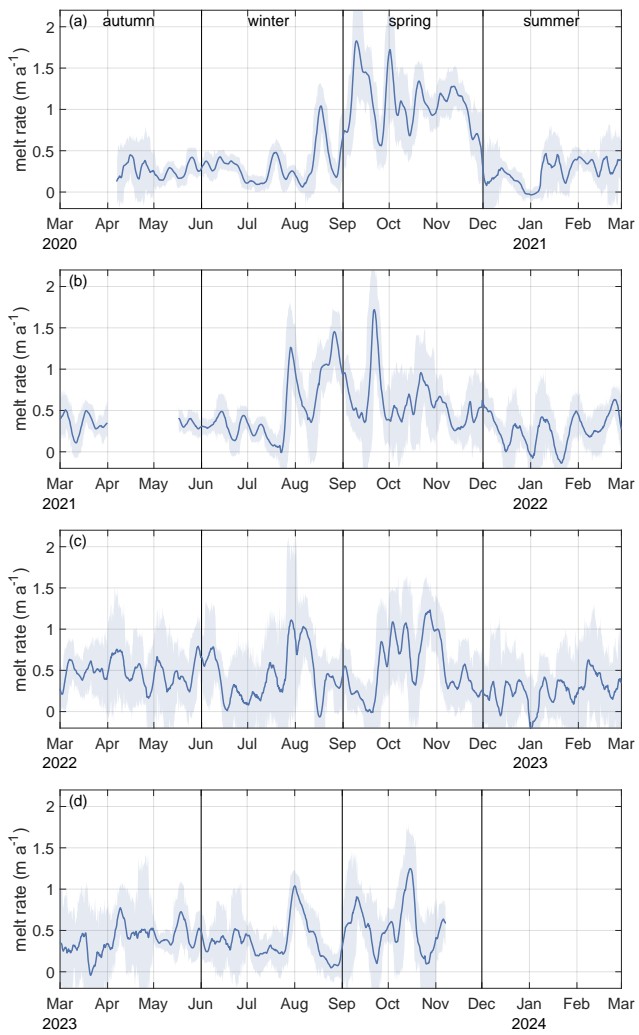

**Figure 2.** Time series of 7-day average basal melt rate from autumn 2020 to spring 2023 (blue line) with sub-weekly variability represented by the standard deviation (shaded area).

## 3.2 Seasonal sea-ice growth and ocean properties

The sea-ice growth in front of the Ekström Ice Shelf exhibits a distinct seasonal pattern, with the lowest growth rates occurring during the summer months and high rates in autumn and spring (Fig. 3b). In late summer and beginning of autumn, when sea-ice concentration is at its lowest, the growth rate quickly increases, exceeding $2\,\mathrm{m\,a^{-1}}$ and leading to a rapid increase in sea ice concentration. When the sea-ice concentration reached a coverage of $90\%$ in April and May, the growth rate initially declines, but increases again towards the end of winter. In October and November, sea ice growth abruptly declines, reaching

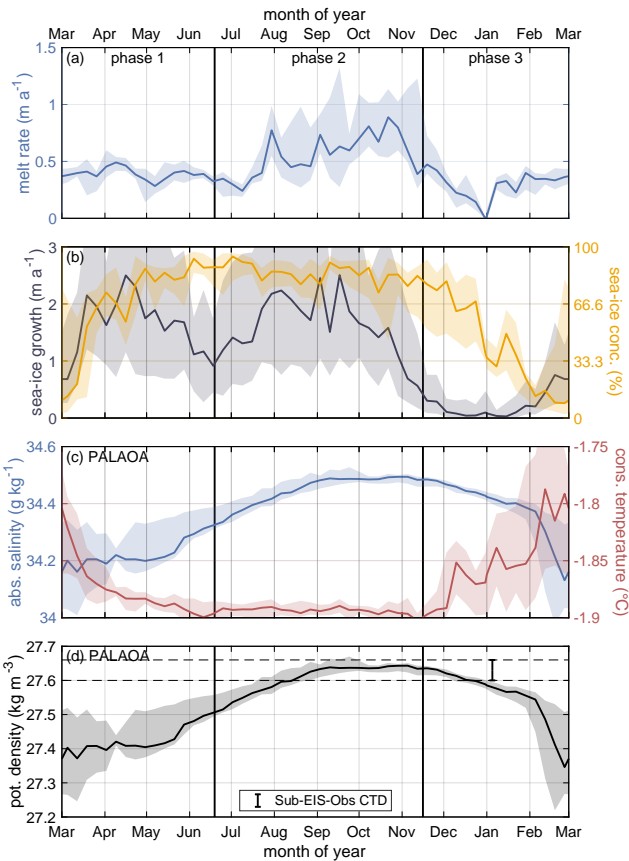

**Figure 3.** Seasonality of basal melt rate, sea-ice growth and concentration, and ocean temperature, salinity and density from autumn to summer. The solid line shows the multi-year average and the shaded area show the $25\%$ to $75\%$ quartile range. The dashed lines show the density range observed with the Sub-EIS-Obs CTD measurements.

values close to zero by December and January. This decline in growth rate is followed by a decrease in sea-ice concentration with a delay of about 2 months.

The full time-series record of sea-ice growth and concentration is shown in Appendix Figures B1 and B2. Similar to the melt rate time series, the high-resolution time series of the sea-ice growth shows an oscillatory variability on weekly time scales that is comparable to the observed basal-melt events. Although some of the melt events can be connected to a previous increase in sea-ice growth, other events in sea-ice growth occurred in autumn without an increase in the melt rate being measured. A cross-correlation analysis between sea-ice formation and melt rate, calculated for each season, revealed predominantly moderate (0.5 — 0.75) or low (0.25 — 0.5) correlation values for various lags of up to $26\,\mathrm{d}$. Particularly, there is no apparent agreement in 2022 and 2023 between the sea-ice growth and the melt rate (Fig. B2).

The seasonalities of ocean temperature and salinity as observed from PALAOA CTD measurements $70\,\mathrm{m}$ beneath the ice base near the ice shelf front are shown in Figure 3c. The temperature exhibits a peak in February at $-1.78\,^{\circ}\mathrm{C}$, followed by a

rapid decline in March and April (Fig. 3c). During winter and spring, the ocean temperatures reached a minimum of $-1.90\,°\mathrm{C}$ and rose again in summer. The salinity reached its lowest value of $33.8\,\mathrm{g\,kg^{-1}}$ in February and March (Fig. 3c). Subsequently, the salinity continuously increased until September and reached its maximum value of $34.5\,\mathrm{g\,kg^{-1}}$ between September and November. This maximum was followed by a slight decrease and a subsequent sharp decline in February. The potential density derived from the combination of temperature and salinity exhibits a similar pattern (Fig. 3d). The highest densities, which exceed the cavity mean, coincide with the enhanced melt rates observed between August and November. Outside of this period, the densities occasionally experience significant reductions, as shown by the lower quartile envelope. The full time-series record of PALAOA temperature, salinity and density is shown in Appendix Figure B3.

## 3.3 Plume model simulations and ISW buoyancy effects on melt rates

The results of the plume model simulations show how the seasonality of varying water masses inside the ice shelf cavity may affect the melt rates below at the Ekström Ice Shelf (Fig. 4). For the seasonal extremes representing autumn and spring, when densest WW is observed at the PALAOA CTD (Fig. 4b,c), a coherent plume ascends from the grounding line to the ice shelf front. This is evident from the smooth evolution of the plume temperature and velocities (Fig. 4d,e), with values exceeding $0.10\,\mathrm{m\,s^{-1}}$ along the entire flow path. In this case, the simulated melt rates at the ApRES location are $1.0\,\mathrm{m\,a^{-1}}$ (Fig. 4f). In contrast, when AASW is present inside the cavity in late summer and autumn, discontinuities in the plume temperature (e.g., approximately $19\,\mathrm{km}$ and $102\,\mathrm{km}$ along the flow path) indicate that the ascending ISW detached and a new plume is initiated locally. The reduced density difference between the ascending ISW and the ambient water results in a reduced plume velocity, averaging $0.06\,\mathrm{m\,s^{-1}}$ (or 57 %) lower than that of the simulation representing the WW case. The decreased flow velocity of the ISW plume reduces the turbulent heat exchange at the ice–ocean interface, leading to a reduced melt rate along the entire flow path and to reduced melt rates of $0.3\,\mathrm{m\,a^{-1}}$ at the ApRES location despite the local temperature increase.

The temperature–salinity diagram (Fig. 4g) compares the seasonal hydrography from the PALAOA site with the Sub-EIS-Obs CTD profiles, illustrating the relative density differences of different water masses inside the cavity. Water that interacts with the ice shelf inside the cavity will undergo a transformation along the meltwater mixing line (Gade, 1979) with a constant ratio of cooling and freshening as it melts the glacier ice. The end-member temperature of this transformation is determined by the pressure-dependent local melting point, which is approximately $-2.05\,°\mathrm{C}$ at the depth of the ice base at the ApRES site.

For water masses with temperatures close to the surface freezing point, as observed in the deeper part of the Sub-EIS-Obs CTD profile, the associated density decrease that may give rise to the meltwater plume is relatively small ($0.06\,\mathrm{kg\,m^{-3}}$) compared to the seasonal density variations that are observed at the PALAOA site ($> 0.2\,\mathrm{kg\,m^{-3}}$ based on the 7 d binned data). Therefore, a vigorous buoyant plume circulation can only develop when winter time convection has eroded the stratification inside the cavity. Even the slight freshening observed at the PALAOA site in December and January (green dots) reduces the density in the upper part of the water column below that of the ascending ISW observed at the Sub-EIS-Obs CTD measurements. In this case, the ISW will not rise but detach from the ice base and mix with ambient water masses of equal density. Additional plume model simulations with an intermediate profile that resembles the Sub-EIS-Obs CTD at depth and transi-

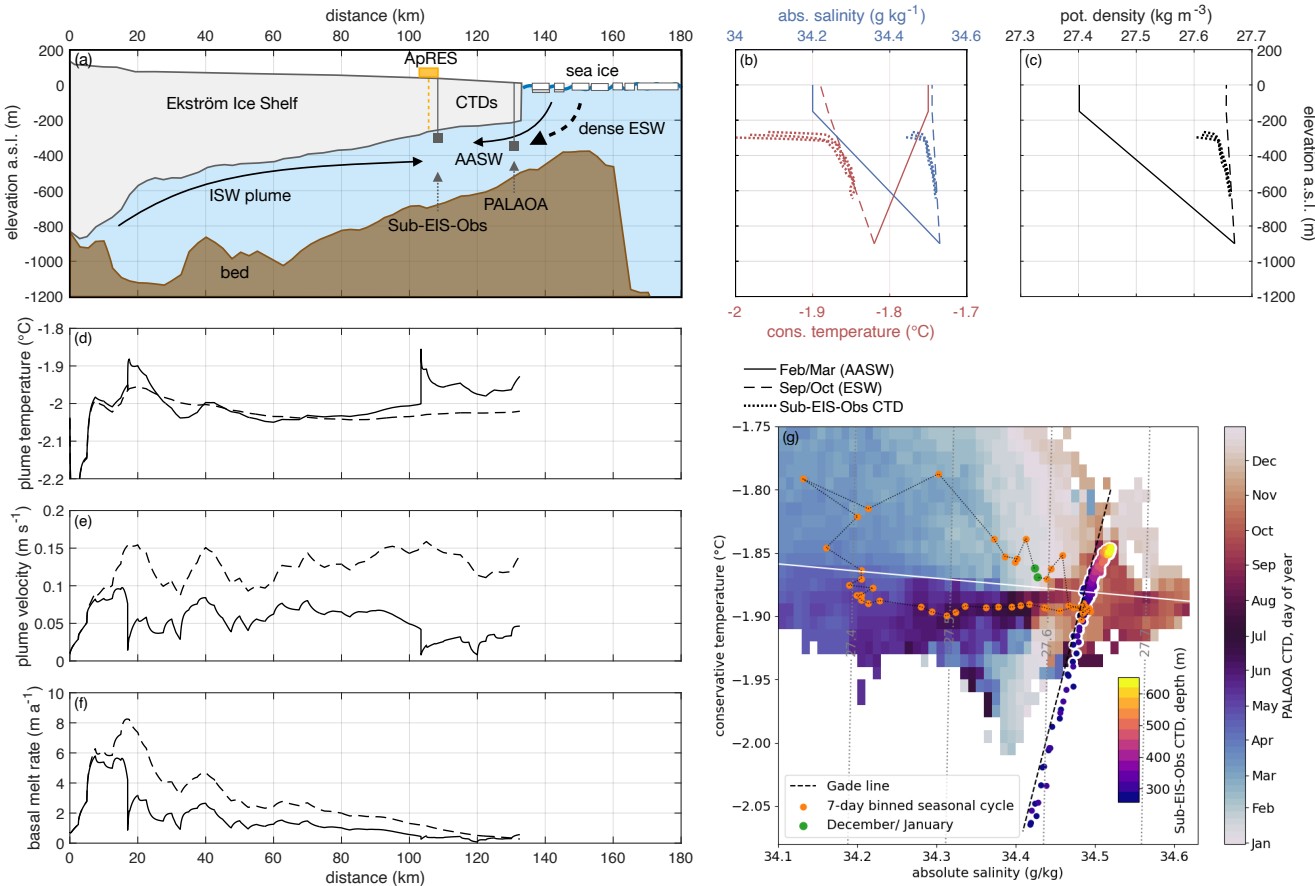

**Figure 4.** Simulation of ISW-plume in the cavity of Ekström ice shelf. (a) Sketch of ice shelf geometry (Morlighem et al., 2020; Morlighem, 2022), bathymetry (Smith et al., 2020) and ocean currents for ISW plume, AASW in summer and ESW in winter. The location of this transect is shown in Figure 1b. (b) Absolute salinity–depth (blue) and conservative temperature–depth (red) profiles for summer (solid lines) and winter (dashed line) conditions, utilized for the plume simulation. Sub-EIS-Obs CTD observations from January 2019 are shown by dotted lines (Smith et al., 2020). (c) Potential density–depth profiles for the given summer and winter conditions and from Sub-EIS-Obs CTD observations. Simulation results for (d) plume temperature, (e) plume velocity, and (f) basal melt rate for summer and winter conditions within the cavity are presented. (g) Conservative temperature–absolute salinity diagram of CTD measurements from PALAOA (2006–2014) and Sub-EIS-Obs (January 2019). The dotted lines show the resulting potential density contours and the white line represents the surface freezing point. The color of PALAOA data (background) represents the seasonality as day of year. The orange dots show the 7 d binned seasonal cycle, with the green dots representing the first week of January and the last week of December. The colour of Sub-EIS-Obs measurements represents the depth below sea level.

tions to the December/January properties observed at the PALAOA site confirm this behaviour, showing reduced melt rates of $0.6\,\mathrm{m\,a^{-1}}$.

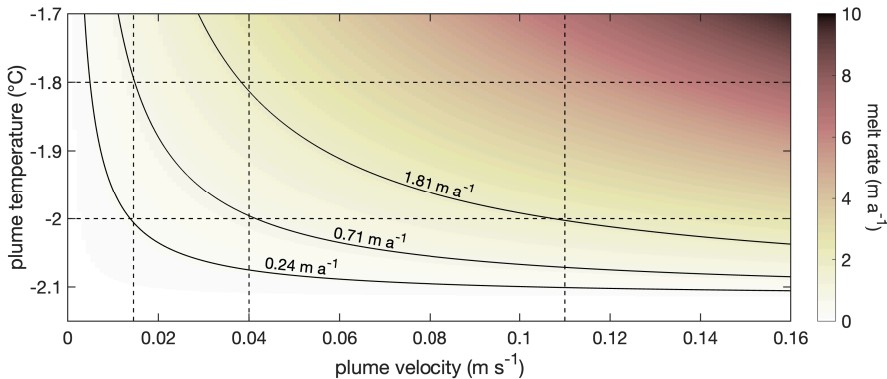

**Figure 5.** Basal melt rate as a function of plume temperature and velocity, calculated using the three-equation model of ice-shelf ocean interaction (Hellmer and Olbers, 1989; Jenkins and Doake, 1991). The solid lines represent the contours for the average melt rate in summer $(0.24\,\mathrm{m\,a^{-1}})$ and spring $(0.71\,\mathrm{m\,a^{-1}})$, as well as the maximum observed melt rate of $1.83\,\mathrm{m\,a^{-1}}$. The horizontal dashed lines denote the grid for plume temperatures of $-1.8$ and $-2\,^{\circ}\mathrm{C}$, while the vertical lines indicate the plume velocities corresponding to the indicated melt rates.

To further evaluate the impact of changes in plume buoyancy, we analyse the influence of ambient temperature and ocean velocity on the modulation of the melt rate, based on the three-equation formulation (Hellmer and Olbers, 1989; Jenkins and Doake, 1991) that is used for parameterising heat fluxes at the ice ocean interface in the plume model. For instance, to achieve a melt rate of $0.24\,\mathrm{m\,a^{-1}}$ that is observed during summer, a plume velocity of $0.015\,\mathrm{m\,s^{-1}}$ is required when assuming a water temperature of $-2\,^{\circ}\mathrm{C}$ (Fig. 5), as observed at the ice base during the early summer of 2018/19 (Smith et al., 2020). Higher melt rates, as are observed during late winter and spring, can be attributed to an increase in plume temperature and/or flow velocity. To obtain the average melt rate in spring of $0.71\,\mathrm{m\,a^{-1}}$, a $0.2\,^{\circ}\mathrm{C}$ higher plume temperature would be necessary to maintain a constant plume velocity. Conversely, an increase in plume velocity to $0.04\,\mathrm{m\,s^{-1}}$ with a constant plume temperature of $-2\,^{\circ}\mathrm{C}$ can also achieve the same melt rate. Both the plume temperature of $-1.8\,^{\circ}\mathrm{C}$ and the velocities of $0.04\,\mathrm{m\,a^{-1}}$ appear to be within a plausible range. However, for a maximum observed melt rate of $1.83\,\mathrm{m\,a^{-1}}$, a higher temperature of up to $-1.3\,^{\circ}\mathrm{C}$ would be required to maintain a constant plume velocity of $0.015\,\mathrm{m\,s^{-1}}$. Such a high plume temperature is not anticipated at the Ekström Ice Shelf, which has the highest weekly median temperatures in the PALAOA CTD around $-1.7\,^{\circ}\mathrm{C}$, during a period of the year when the observed melt rates are lowest. If we assume that an increase in temperature alone was not the sole cause of the high melt rate, then the flow velocity of the plume must have also increased. At a constant temperature of $-2\,^{\circ}\mathrm{C}$, an increase in the flow velocity to $0.11\,\mathrm{m\,s^{-1}}$ could have also resulted in a melt rate of $1.83\,\mathrm{m\,a^{-1}}$, which still falls within a plausible range. This analysis suggests that the increase in melt rates in winter and spring cannot be attributed solely by warmer water, but rather to higher plume velocities.

## 4 Discussion

Melt rates determined from in situ data, such as those derived from ApRES measurements, have a high accuracy and can therefore be used to validate melt rates estimated from satellite measurements. However, a detailed comparison of the melt rates determined from ApRES measurements in this study with those estimated from satellite observations is difficult due to different observational periods. The melt rates determined from Adusumilli et al. (2020) at the ApRES site for the period 2010 to 2018 are $0.91 \pm 0.82 \, \mathrm{m \, a^{-1}}$, which is higher than the ApRES-derived melt rates from 2020 to 2023. Detailed comparisons of the spatial distribution and temporal variability between ApRES measurements on the Filchner-Ronne Ice Shelf with the results from Adusumilli et al. (2020) have revealed large differences, which are due to uncertainties in the satellite-based method (Vaňková and Nicholls, 2022; Zeising et al., 2022). The latest satellite-based melt rate estimates from Davison et al. (2023) for the period 2010 to 2021 show lower melt rates of $0.65 \pm 0.13 \, \mathrm{m \, a^{-1}}$ near the ApRES site, which are still above the annual mean values derived from the ApRES data.

At the Nivl ice shelf, Lindbäck et al. (2019) observed higher melt rates of up to $5.6 \, \mathrm{m \, a^{-1}}$ when warm AASW is pushed beneath the ice shelf by winds. However, at greater distances of $35 \, \mathrm{km}$ from the ice front, Lindbäck et al. (2019) detected no pronounced seasonality and lower melt rates of below $2.0 \, \mathrm{m \, a^{-1}}$ despite a single melt event in winter. While it is plausible that downwelling of AASW during summer results in mode 3 melting near the ice front also beneath the Ekström Ice Shelf, the melt rate seasonality at the ApRES location approximately $30 \, \mathrm{km}$ from the ice shelf front at an ice draft depth of $\sim 250 \, \mathrm{m}$ indicates a different effect of the seasonally varying hydrography. Ice shelf cavity moorings at the Fimbulisen Ice Shelf Observatory (Lauber et al., 2024) showed seasonal traces of AASW as far as $60 \, \mathrm{km}$ from the ice shelf front and below $380 \, \mathrm{m}$ deep ice. At this location, the AASW had depleted all of its additional heat for melting and primarily affected the cavity circulation by separating the ISW from the ice base. A similar mechanism is inferred for the ApRES location below the Ekström Ice Shelf. In contrast to Fimbul and the Nivl Ice Shelf, the Ekström Ice Shelf also exhibits a more confined geometry and a deeper grounding line, which together promote the formation of a coherent cavity overturning circulation that is primarily driven by the pressure-dependent formation of buoyant ISW. The interaction of this circulation with the seasonally varying upper ocean stratification appears to be the dominant driver of seasonal melt rate variability at the ApRES location.

We propose that convective mixing of dense water that is generated during sea-ice formation in late winter and spring enhances the buoyancy contrast and accelerates the flow of ascending ice shelf water. The convective mixing results in a homogenous water mass inside the cavity with properties similar to those of the lower portion of the Sub-EIS-Obs CTD profile. In this scenario, the meltwater input provides sufficient buoyancy for the plume to rise within the homogenous water column to the ice shelf front, maintaining substantial flow velocities that facilitate shear-driven melting at the ApRES site. The absence of the dense water formation and a restratification of the water column during the summer and autumn reduces the buoyancy of the ascending ISW, causing the arrest of the buoyant plume as it encounters water masses of the same density at the ApRES site. Conversely, the erosion of the stratification at the onset of convective mixing will potentially revitalise the plume-type circulation. This interpretation is supported by the plume model, which demonstrates how the seasonal extremes in temperature and salinity as observed with CTD measurements, affect the plume velocity and thus the basal melt rate.

Based on the aforementioned findings, we propose the following hypothesis of the sea ice-ice shelf-ocean interaction, which we divide into three phases (Fig. 3):

1. Phase of sea-ice regrowth (March to June): During the autumn and early winter months, the AASW cools to its freezing point. The initial sea-ice production over the open ocean is substantial until the regrown ice cover reduces further heat loss from the ocean. The remaining stratification within the cavity generates a low-density contrast between the rising ISW plume and the ambient water. The reduced density contrast diminishes the buoyancy and velocity of the meltwater plume, resulting in reduced melt rates.

2. Phase of convection (June to November): Cold mid-winter atmospheric temperatures and sustained sea-ice growth induce brine release, which convectively mixes and densifies the water column in front of the ice shelf. Once the surface density exceeds a certain threshold, it erodes the stratification on the continental shelf and descends along the seafloor towards the grounding line inside the ice shelf cavity. The observed density range of the Sub-EIS-Obs CTD profiles indicates that convection may occur during the period when enhanced melt rates are observed (Fig. 3). During this period, the density contrast to the ambient water accelerates the ISW plume, thereby enhancing the turbulent heat flux and melting at the ice-ocean interface. At the beginning of Winter, before dense water inflows have eroded the stratification within the entire cavity, transient fluctuations that enhance the density contrast for a finite duration, are causing the observed melt events.

3. Phase of restratification (November to February): The onset of sea-ice melt in November and December, coupled with melting at the ice shelf front due to increased solar radiation, leads to a reduction in the upper ocean density. This causes the plumes to separate from the ice base, resulting in an abrupt decline in melt rates. In January and February, sustained accumulation and downwelling of AASW (Zhou et al., 2014) continue to freshen and stratify the continental shelf water masses. The stratification within the cavity generates a low-density contrast between the rising ISW and the ambient water. This reduced density contrast diminishes the buoyancy and velocity of the meltwater plume, leading to low melt rates in summer and autumn.

In the hypothesis presented, the formation of dense water through sea-ice formation emerges as a driver for the erosion of stratification. The direct comparison between sea-ice growth and melt rate revealed moderate correlations. Although a consistent temporal lag was not evident, an estimated advection time scale of $8$ to $16\,\text{d}$ from the ice front to the grounding line may well explain the observed approximate phase shift between the sea-ice formation and the melt rate response, as the dense water signal propagates through the cavity.

While the observed water mass evolution at the PALAOA site supports our hypothesis of the interaction of the buoyancy-driven cavity overturning circulation with the seasonal changes in upper ocean stratification, the plume model simulations only provide a simplified representation of the dynamics at play. In reality, the one-dimensional overturning captured by this model will be affected by rotation at the cavity scale, where the ISW flow would be deflected to the left to follow geostrophic contours and rise where it is constrained by topography to flow to upslope topography (Holland and Feltham, 2006). A three-dimensional circulation model of the Fimbul Ice Shelf cavity (Hattermann et al., 2014) shows that such constraints may be provided by a central keel that guides an upward flow of ISW along its eastern flank. A similar keel-like topographic feature is observed along the central flowline of the Ekström Ice Shelf (Fig. 1b), and the ApRES is located along its eastern flank. Running the plume model with various (idealised) geometries confirms the robustness of results about the influence of the cavity stratification in

the simulated melt rates regardless of choice of the detailed flow path of the plume. Another component that is not represented by the plume model is the estuarine-like aspect of the the cavity overturning circulation that relates the strength of dense water inflows at depth and buoyant outflows below the ice base through continuity (Olbers and Hellmer, 2010), or any other effects of the time varying ambient circulation that may play a role for shaping the observed melt rate oscillations. The plume model also does not explicitly capture tidal effects, such as the increase of entrainment of ambient water into the plume or turbulent mixing at the ice-ocean interface. However, since these effects superimpose on the simulated processes, we assume that the proposed impact of seasonal stratification on the plume dynamics is a robust result, while the detailed modulation of these dynamics remains subject for further studies.

The existing melt rate measurements in DML as well as the plume simulation show that the melt rate as well as the intensity and type of seasonality can strongly depend on the relative position on the ice shelf. Further measurements are needed to better understand the spatial and temporal variation of the basal melt rates. In addition, ocean simulations indicate that an increase in melt rates of the eastern Weddell Sea ice shelves can trigger a melt regime change of Filchner-Ronne Ice Shelf ($14\,\mathrm{m}$ sea-level equivalent (Rignot et al., 2019)) towards significant higher melt rates (Hoffman et al., 2024). For this reason, melt rates in DML and of other ice shelves in the Weddell Sea should be further monitored using a network of ApRES.

## 5   Conclusions

In this study, we investigated the ice–ocean interaction at the Ekström Ice Shelf using a combination of basal melt rate observations from a 4-year-long ApRES time series, a time series of sea-ice and ocean properties, as well as an ice shelf plume simulation. Our analysis provides valuable insights into the seasonality of basal melting, contributing to a deeper understanding of the processes driving ice shelf basal melting in East Antarctica. The melt rates were lowest during the summer and autumn and highest in the spring, with a few weeks-long melt events occurring predominantly from late winter to early spring. An investigation of the seasonality of sea-ice growth and ocean density highlights the impact of sea-ice formation on ocean conditions in the ice-shelf cavity. Results from an idealised meltwater plume model are consistent with the hypothesis that melt rates decrease in summer and autumn because enhanced stratification inside the cavity weakens the buoyancy driven cavity circulation. Due to low contrast in density, the velocity of the ice shelf water plume reduces and, as a result, the melt rates decrease. In spring, the plume velocity is enhanced due to dense water from the sea-ice formation that erodes the stratification and increases the density contrast. We conclude that the melt rate variability is primarily affected by the plume velocity, while the temperature of the ambient water is only secondary. Overall, our study underscores the importance of continuous and high-resolution observations to capture the temporal variability of basal melt processes. The findings emphasise the need for further research to analyse the processes under the entire ice shelf as well as of other ice shelves of the Dronning Maud Land to correlate melt rate trends with a climate signal. In addition, the use of models that can capture complex processes is needed, as well as oceanic measurements to validate model results. By enhancing our understanding of these ice–ocean interactions, we can better assess the future contributions of Antarctic ice shelves to global sea-level rise.

*Data availability.* Time series of autonomous phase-sensitive radar (ApRES) measurements and determined basal melt rates are published at the World Data Center PANGAEA (https://doi.org/10.1594/PANGAEA.972121, Zeising et al., 2024). Time-series of conservative temperature, absolute salinity and potential density from PALAOA CTD measurements are published at XXX.

## Appendix A: ApRES analysis

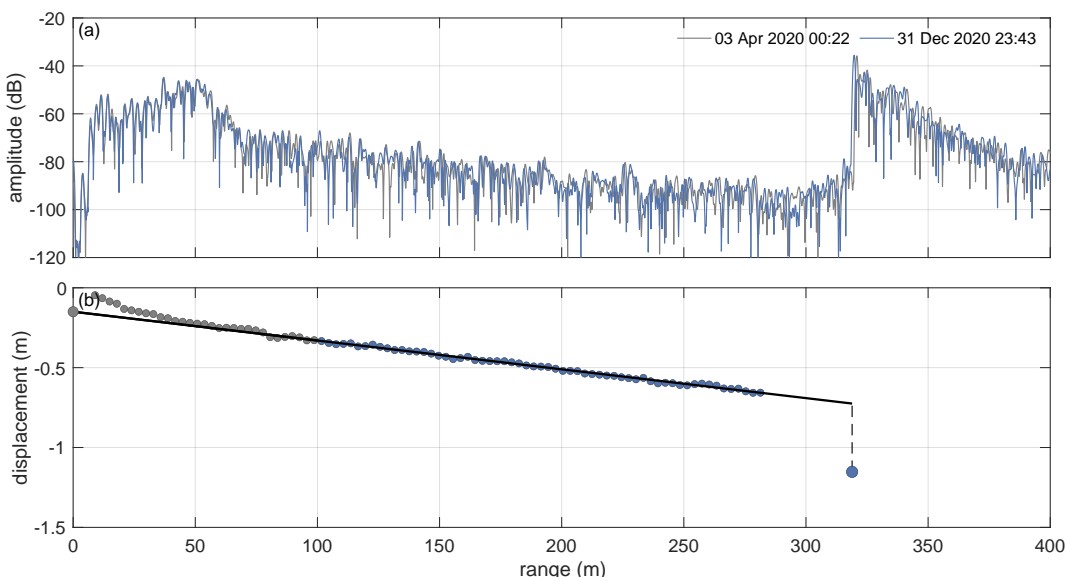

**Figure A1.** Basal melt rate analysis of ApRES data from 2020. (a) Amplitude profiles of the first (grey line) and last measurement (blue line) in 2020. (b) Vertical displacement of the basal segment (large blue dot) and of internal segments (small dots). Those internal segments used for strain rate analysis (gradient of solid black line) are coloured in blue. The dashed line represents the displacement caused by basal melting.

**Table A1.** Derived quantities from ApRES measurements (03 April 2020 – 31 December 2020; Fig. A1) used for basal melt rate determination: Ice thickness change ($\Delta H$), thickness change due to firn compaction ($\Delta H_f$), due to vertical strain ($\Delta H_\varepsilon$), the vertical strain itself ($\dot{\varepsilon}_{zz}$) and the basal melt rate ($a_b$). Negative values contribute to the thinning of the ice column, whereas a positive melt rate represents melting.

| Year | $\Delta H\,(\mathrm{m\,a^{-1}})$ | $\Delta H_f\,(\mathrm{m\,a^{-1}})$ | $\dot{\varepsilon}_{zz}\,(\mathrm{a^{-1}})$ | $\Delta H_\varepsilon\,(\mathrm{m\,a^{-1}})$ | $a_b\,(\mathrm{m\,a^{-1}})$ |
|------|------|------|------|------|------|
| 2020 | $-1.54\pm0.01$ | $-0.20\pm0.01$ | $-0.0024\pm0.0001$ | $-0.77\pm0.01$ | $0.57\pm0.01$ |

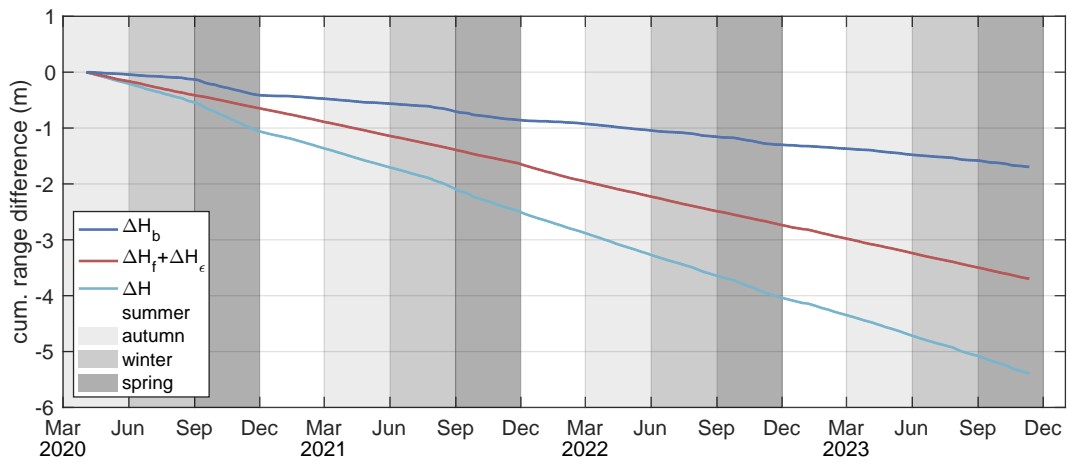

**Figure A2.** Cumulative change in ice thickness ($\Delta H$), attributed to strain and firn compaction ($\Delta H_f + \Delta H_\varepsilon$) as well as basal melting ($\Delta H_b$) from April 2020 to November 2023. The vertical stripes denote the respective seasons.

 **Appendix B: Sea-ice and CTD time-series**

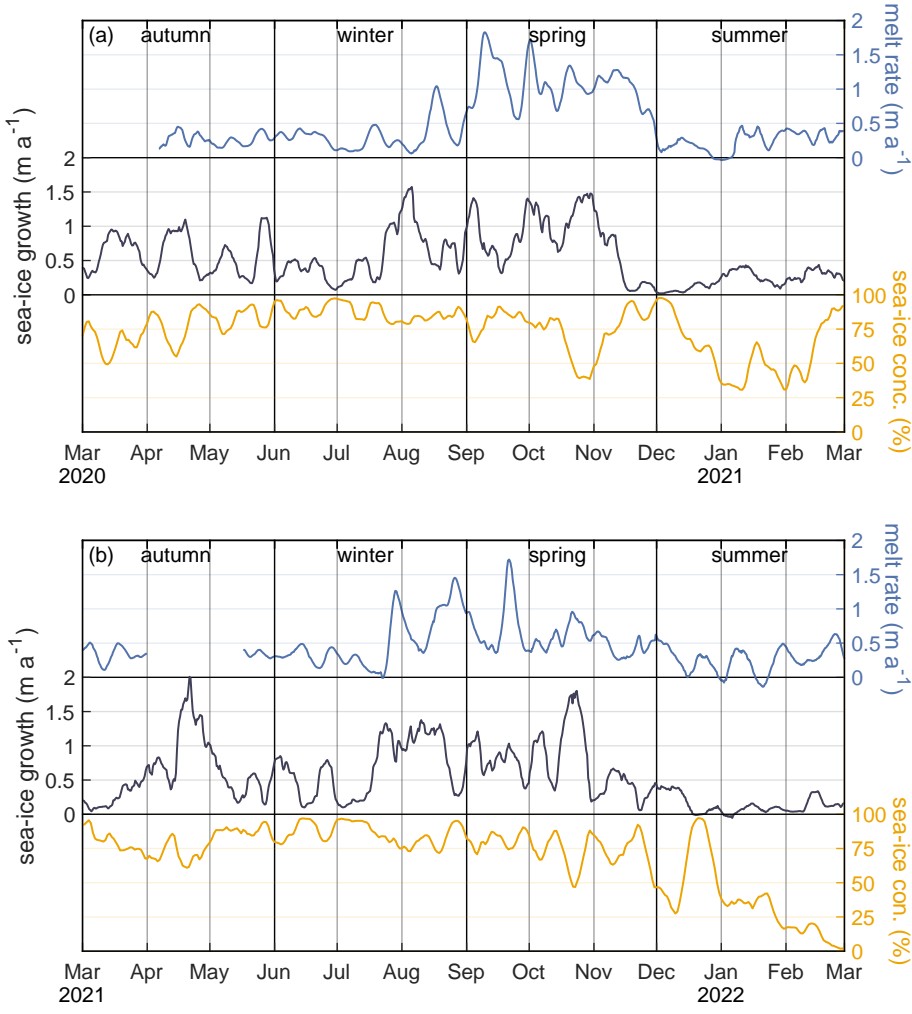

**Figure B1.** Average sea-ice growth and concentration (black lines) above continental shelf (Fig. 1b) and basal melt rate (blue line) from 2020/21 to 2021/22.

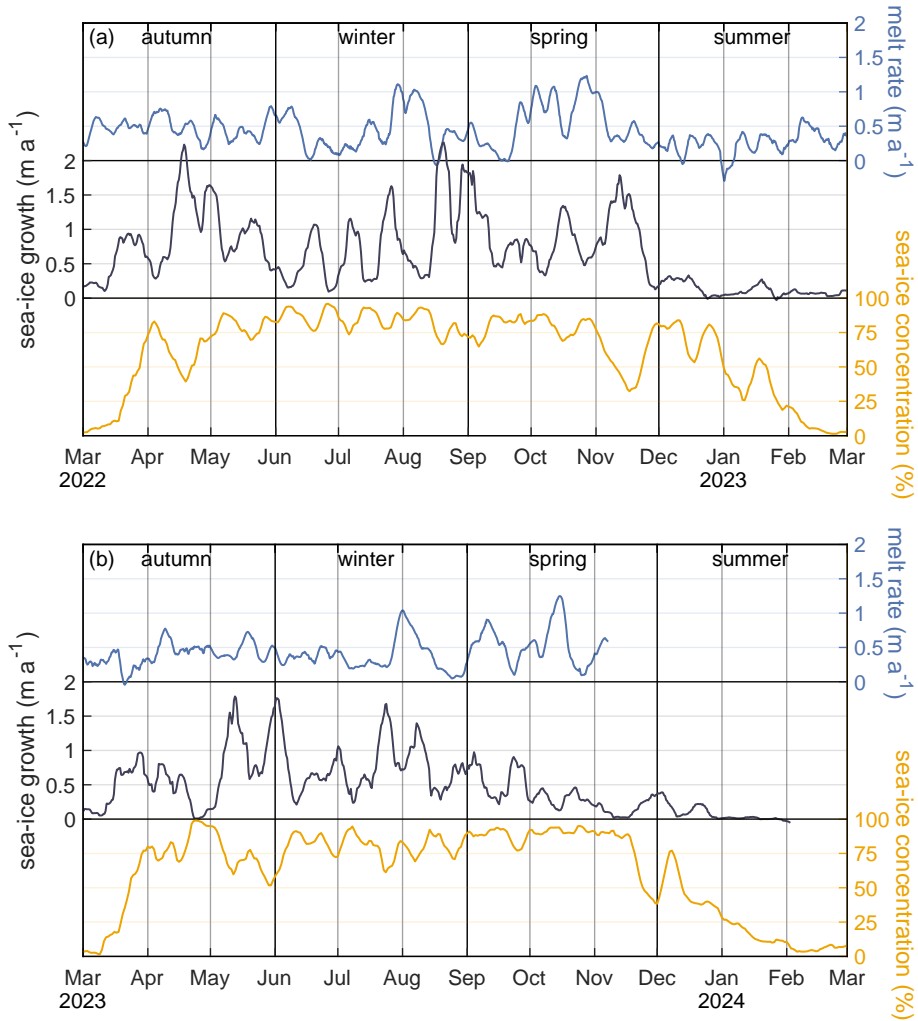

**Figure B2.** Average sea-ice growth and concentration (black lines) above continental shelf (Fig. 1b) and basal melt rate (blue line) from 2022/23 to 2023/24.

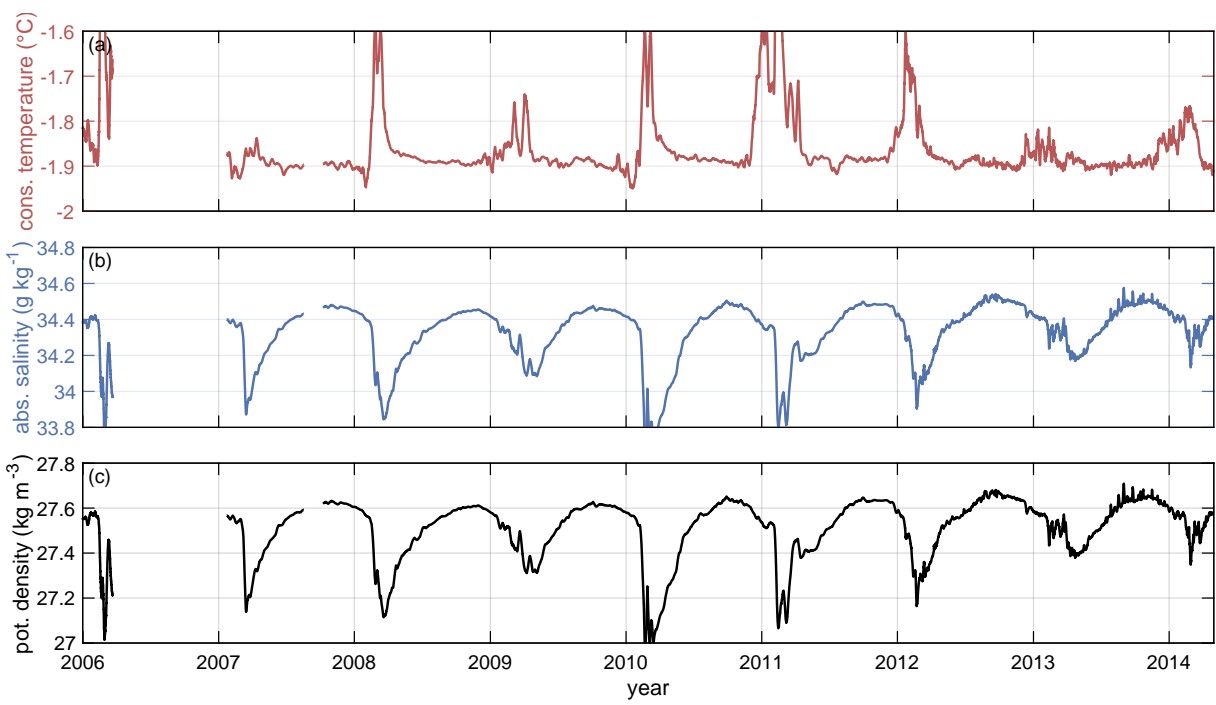

**Figure B3.** Time-series of conservative temperature, absolute salinity and potential density from PALAOA CTD measurements.

*Author contributions.* OZ and OE designed the study. OE initiated the long-term observations, with support from FP and RD, and implemented the logistic operations. DS and OZ assembled the hardware components. Measurements were performed by SB, MRE and OE in multiple years. TF guided the overwinterers as part of the Geophysical Observatory. OZ processed the ApRES data and wrote the manuscript with contributions from TH and LK. TH performed the plume simulations, and analysed the seasonality. LK processed the satellite data and determined the sea-ice growth and concentration. OB and TH processed the CTD data from PALAOA. OE, RD, TH and LK contributed to data discussion. All authors contributed to revising the manuscript.

*Competing interests.* At least one of the (co-)authors is a member of the editorial board of The Cryosphere.

*Acknowledgements.* The authors wish to thank the overwinters and summer guests at the German research station Neumayer III (Alfred-Wegener-Institut Helmholtz-Zentrum für Polar- und Meeresforschung, 2016) who maintained the ApRES once a season: Noah Trumpik, Ina Wehner (2019/20 & 2020/21); Timo Dornhöfer, Lorenz Marten (2020/21 & 2021/22); Falk Oraschewski, Martin Petri, Maximilian Betz (2022/23); Hameed Moqadam, Fyntan Shaw (2023/24). We thank Ralph Timmermann for discussion in an early stage of this study.

Financial Support

Initial funding was provided through the grant the Belgian Research Programme on the Antarctic (Belgian Federal Science Policy Office) through the BELSPO MIMO project (Stereo III). The project MIMO-EIS in the framework of the quasi-Long-Term-Observatory Glaciology at Neumayer station III are supported by the Alfred-Wegener-Institut Helmholtz-Zentrum für Polar- und Meeresforschung (AWI), project GLAZ-NM (Grant-No. AWI_ANT_8) and project ReMeltRadar (Grant-No. AWI_ANT_18). R.D. and M.R.E were supported by a grant of the Deutsche Forschungsgemeinschaft (Grant-No. DR 822/3-1). T.H. was supported by the Research Council of Norway through the Centre for ice, Cryosphere, Carbon and Climate, project 332635, and I-CRYME, project 2560594.

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
