# Peer review of "Enhanced basal melting in winter and spring: Seasonal ice-ocean interactions at the Ekström Ice Shelf, East Antarctica"

_EGUsphere, 2024_

## Referee Comment (RC2)

[referee-annotated manuscript omitted]

---

## Author Response (AR1)

Dear Reviewers,

We would like to thank you again for your efforts to improve our manuscript. We acknowledge the concerns you have raised and for the thorough assessment of the dynamical hypothesis that we put forward to explain the observed melt rate seasonality. In the revised version of the manuscript, we made substantial textual changes, added new figures and changed the title. We acknowledge more carefully and discuss that the plume model we employed to investigate the melt rate variability, remains somewhat simplified. At the same time, your constructive feedback has led us to additional analysis of available hydrographic data that is available to assess the seasonal variability and interplay of different water masses inside the Ekström Ice Shelf cavity, which provided additional evidence to strengthen our hypothesis of the dynamical process that appears to strengthen (reduce) basal melting in late winter and fall (late summer and autumn) at the ApRES site. Together with a clarification of the nomenclature of the seasonal extremes that were presented in the previous version of the manuscript, we are confident that the revised version of the manuscript provides a better understanding of the processes that ice-ocean interactions of the Ekström Ice Shelf.

Thank you once again for your time and expertise.

Kind regards,
Ole Zeising and co-authors

**Authors point-to-point response on Referee Comment #1 to egusphere-2024-2109**

This study poses a novel mechanism underlying seasonal variations in the melt of Ekström Ice Shelf, linking it to seasonal variations in sea ice production, densification and destratification of the waters in the ice shelf cavity, and subsequent acceleration of the meltwater plume. To achieve this, the authors present a novel estimate of melt rates in the center of Ekström Ice Shelf with fine temporal resolution using an autonomous phase-sensitive radar system. They compare this with time series of sea ice formation rates estimated using satellite derived open water area, a reanalysis product, and bulk formulae to estimate ocean-air heat fluxes. This resulting seasonal variations in sea ice production and ice shelf melt rates approximately co-vary, with sea ice formation appearing to slightly lead the melt rates. This leads the authors to pose the hypothesis that variations in sea ice lead to changes in the density of the waters in the EIS cavity, with a higher density contrast in winter that accelerates the melt plume and thus increase the melt rates.

This topic is certainly relevant to, and of interest to readers of, The Cryosphere, and the manuscript is well-written and clear (if rather terse in places). The portion of the manuscript describing the melt rate time series is detailed, with novel results that are a valuable contribution to ongoing community efforts to quantify and understand basal melt of ice shelves. However, the portions of the manuscript that link the melt rates to sea ice production and melt plume dynamics are less convincing, and remain plausible but somewhat speculative, despite the evidence presented. To elaborate on this point:

1. The link between the plume velocity and sea ice formation is emphasized in the abstract and discussion, but is not strongly supported by the evidence presented in the manuscript. The seasonal cycles of sea ice formation and ice shelf melt rates line up nicely, but does not imply a causative relationship. Furthermore, on shorter time scales there is no clear link between sea ice formation and melt rates. The authors' plume simulations are certainly consistent with the idea that variations in sea ice formation could lead to changes in melt plume velocity, but involve the assumption that the changes in cavity stratification result only from changes in sea ice production (rather than lateral advection, for example), in addition to various other idealizations in the plume model that the authors have noted.

2. The plume modeling comprises a relatively minor portion of the manuscript, yet seems to have developed into the central focus of the paper: for example, the title focuses specifically on the meltwater plume as the driver of the melt. This contrasts with the focus of the manuscript itself, which is centered on observations of melt rates.

In summary, the authors have produced a manuscript whose primary result is a novel observational estimate of basal melt rates, complemented by a plausible hypothesis that the seasonal cycle of the melt rates is driven by sea ice formation-modulated variations in melt plume velocity. Yet somehow the plausible hypothesis has acquired central emphasis in the title, abstract and conclusion, despite limited evidence. I recommend that the manuscript be returned to the authors for major revisions so that they can either re-frame their findings or more strongly support their claims.

Other comments/questions:

L. 1–2: "Basal melting of ice shelves significantly contributes to the mass loss of the Antarctic Ice Sheet. However, little is known about the ocean-driven melting of the numerous ice shelves of Dronning Maud Land in East Antarctica."

Reviewer:
- This statement of the gap in understanding is overly vague and general: there is some previously-established understanding of ocean-driven melting in this region, and the abstract could be more specific about placing the work in this context.

Authors' response:
We agree that the beginning of the abstract was not specific enough. In the revised version, the new introduction of the abstract reads as follows:

L. 1–4: *"Basal melting of Antarctic ice shelves significantly contributes to ice sheet mass loss, with distinct regional disparities in melt rates driven by ocean properties. In East Antarctica, cold water predominantly fills the ice shelf cavities, resulting in generally low annual melt rates, as observed in Dronning Maud Land (DML). In this study, we present a four-year record of basal melt rates at the Ekström Ice Shelf, measured using an autonomous phase-sensitive radar (ApRES)."*

L. 91–94: "Estimated basal melt rates of the Ekström Ice Shelf from satellite remote sensing methods are relatively low. Adusumilli et al. (2020) found an average basal melt rate of 1.0±1.2 m/a (period 1994 – 2018) over the entire ice shelf and a maximum melt rate exceeding 3 m/a near the grounding line as well as near the German station Neumayer III. Slightly lower melt rates of <1.1 m/a were found by Neckel et al. (2012) for the period between 1996 and 2006."

Reviewer:
- These previous estimates are only given a cursory mention in the manuscript, yet the authors' novel time series of basal melt is one of their central results, and so warrants more extensive discussion in the context of previous work. Here, or in the discussion section, the authors should discuss the differences between their melt rate estimates vs. previously published estimates, and possible reasons for the differences between them. A key question on readers' minds is going to be how much confidence to place in these versus previous estimates, and any information the authors can provide to help readers make that judgment will improve the manuscript.

Authors' response:
Thanks for raising this point. The advantage of ApRES measurements is that all processes that contribute to the change in ice thickness can be derived in high accuracy. This is different for satellite-remote sensing-based melt rates. These are based on different (simulated) products like strain rate from surface velocities, accumulation, firn-air content and surface height that needs to be corrected for tides. Thus, ApRES measurements are more accurate than satellite-based products. We added the following text to the discussion:

L. 326–335: *"Melt rates determined from in situ data, such as those derived from ApRES measurements, have a high accuracy and can therefore be used to validate melt rates estimated from satellite measurements. However, a detailed comparison of the melt rates determined from ApRES measurements in this study with those estimated from satellite observations is difficult due to different observational periods. The melt rates determined from Adusumilli et al. (2020) at the ApRES site for the period 2010 to 2018 are 0.91±0.82 m/a, which is higher than the ApRES-derived melt rates from 2020 to 2023. Detailed comparisons of the spatial distribution and temporal variability between ApRES measurements on the Filchner-Ronne Ice Shelf with the results from Adusumilli et al. (2020) have revealed large differences, which are due to uncertainties in the satellite-based method (Vanková and Nicholls, 2022; Zeising et al., 2022). The latest satellite-based melt rate estimates from Davison et al. (2023) for the period 2010 to 2021 show lower melt rates of 0.65±0.13 m/a near the ApRES site, which are still above the annual mean values derived from the ApRES data."*

L. 101: "The received signal is mixed with a replica of the transmitted signal to obtain a de-ramped signal that is sampled with 40kHz (Nicholls et al., 2015)."

Reviewer:
- I found "mixed with a replica of the transmitted signal" to be difficult to interpret: could the authors please explain this procedure more precisely?
- Also I was also confused by the 40kHz sampling. This may simply stem from a fundamental misunderstanding of the analysis on my part, but isn't this sampling period much longer than the two-way travel time through the ice (based on the propagation speed given later)? If so, how do the authors resolve distances down to 1mm?

Authors' response:
The ApRES is a FMCW (frequency modulated continuous wave) radar that transmits and receives a signal at the same time. Because the frequency of the transmitted signal increases with a known rate (200 to 400 MHz within 1 s), the two-way travel time of a reflection can be determined from the frequency difference between the received signal and the contemporaneous transmitted signal. The signal that contains this frequency difference is the de-ramped signal that is obtained from "mixing" the received and contemporaneous transmitted signal by the ApRES. The return amplitude and phase as a function of two-way travel time can be calculated from a fourier transformation of the de-ramped signal. Thus, a sampling frequency of 40 kHz is sufficient to determine the frequencies of the mixed signal with sufficient accuracy. The high resolution of up to 1 mm in the vertical displacement is achieved by phase differences from two measurements at different times.

We understand that the sentence used in the manuscript requires in-depth knowledge of radar technology, but it is not necessary to understand the method used. Therefore, we will remove the sentence from the manuscript and refer the reader to the original publications introducing pRES and its operation.

Fig. 1

Reviewer:
- Might it be helpful to include a map of previously-reported melt rates, e.g. those of Adusumilli et al.? In addition to providing a clearer point of comparison, this would also help readers to judge how representative the melt variations at ApRES are of other parts of the ice shelf.

Authors' response:
We agree that showing the satellite remote sensing-based melt rates from Adusumilli et al., (2020) would help to quantify spatial variability and thus assess how representative the annual mean melt rate derived from ApRES is. However, the melt rate from Adusumilli et al. (2020) was measured over a different time period. Still, we added the melt rate distribution to Fig. 1c and we emphasize the need for the ApRES measurements to evaluate these satellite-based products.

L. 97-115: Methods: ApRES

Reviewer:
- I found the description of the methodology rather hard to follow. I think the issue is that it is written with enough detail to summarize the key steps, but not enough detail for a reader to completely reproduce all of those steps. If this methodology is standard then some of the detail could be omitted and replaced with appropriate references. If not then the authors need to be more precise here; I would recommend expressing some of the key concepts as equations, and perhaps including a figure illustrating the analysis procedure.
- This portion of the text suffers from a similar issue as above: the ideas are well articulated, but are difficult to follow without equations that precisely express the calculations and figures that illustrate the process.

Authors' response:
We understand that the description of the methodology is not easy to follow. We have already provided several references where the method is also described. We have described the methodology with the aim that someone with ApRES experience can reproduce the processing, especially the small deviations from approaches used in other studies. We have slightly shortened the description of the signal processing to enhance readability. Not all details and equations can be listed in this manuscript, which is why we refer to the other studies here. Two figures (one of them is new) in Appendix A illustrate the method of melt rate analysis we applied here.

L. 109-110: "Next, to obtain an amplitude and phase profile from the stacked de-ramped signal, we applied a spectral analysis."

Reviewer:
- "Spectral analysis" is rather vague. It sounds like the authors may be computing overlapping windowed spectra, but I am really not sure and would appreciate a more thorough description of the approach here.

Authors' response:
We agreed and changed "spectral analysis" to "Fourier transformation", which is what we do here.

L. 114: "Finally, we obtained a range profile by converting the two-way travel time into a range using a propagation velocity of 168914 km/s according to the ordinary relative permittivity of 3.15."

Reviewer:
- Please provide citations for the propagation speed and relative permittivity.

Authors' response:
We added a citation for the propagation velocity and relative permittivity.

L. 144: "[...] where the firn compaction $\Delta Hf$ is the intercept at the surface."

Reviewer:
- How does surface melt factor into the ice thickness change budget?

Authors' response:
The analysis of the vertical displacements within the ice would also include surface melting if this has happened beneath the antennas. Since the antennas were more than half a meter below the surface, potential surface melting (which is rare anyway) did not affect the measurements at Ekström Ice Shelf.

L. 155: "On average the uncertainty of the basal melt rate amounts to 0.02 m/a."; Eq. (4)

Reviewer:
- Please briefly explain how this uncertainty was computed.
- How accurate is the assumption of constant strain close to the upper and lower surfaces of the ice shelf? (This is somewhat out of my area of expertise.)

Authors' response:
Based on the vertical displacements we determined with the ApRES measurement, we obtained a depth profile of displacements from near the surface (the upper 10 to 15 m cannot be analysed) to about 90% of the ice thickness. In the upper 100 m, the displacement is influenced by firn compaction and was thus not used for strain rate analysis. From 100 m

to roughly 280 m (~90% of ice thickness) the vertical displacements are sufficiently represented by a constant strain (linear increase in displacement over depth, see Appendix). Thus, the only assumption we made here is that the strain in the last 10% of the ice thickness (where we cannot determine the displacement) is the same as between 30% and 90% of the ice thickness. Since the measurement location was on an ice shelf and additionally is not influenced by a subglacial channel, it is accurate to assume that the strain rate is constant in the last 10% of the ice thickness.

In the revised version, we added the following to the uncertainty analysis:
L. 158–165: *"The uncertainty of the vertical displacements are based on the standard error of the radar phase which is in the order of sub-millimetres. The largest uncertainty originates from the strain analysis and its extrapolation to the ice base. The ApRES measurements at Ekström Ice Shelf allow the determination of the depth profile of displacements from near the surface to about 90% of the ice thickness. The only assumption we made here is that the strain in the lowest 10% of the ice thickness is the same as above. Since the measurement location was on a freely floating part of an ice shelf, the vertical displacements are sufficiently represented by a constant strain. Thus, it is accurate to assume that the strain remains constant in the remaining 10% of the ice thickness. On average the uncertainty of the basal melt rate amounts to ~0.02 m/a."*

Section 3.2: Sea-ice growth

Reviewer:
- This approach is similar to that of Tamura et al. (2016), but there isn't any discussion of how this method compares (in approach and in result) to previous estimates of sea ice formation. Please expand this section (and perhaps the discussion section) to place this part of the work in the context of previous studies.

Authors' response:
The approach used in this study is similar to that of Tamura et al. (2016) and Oshima et al. (2016) in that it employs a heat budget (HB) method. However, there are key differences in the methodology and the satellite data applied. Lin et al. (2023) compared sea-ice production (SIP) estimates derived from two different methods: the HB method and the sea-ice volume conservation (VC) method. The VC method requires high-resolution and accurate sea-ice velocity data, which is unavailable for our region of investigation.

Lin et al. (2023) also highlighted a major limitation of the HB method: the unrealistic assumption of zero ocean heat flux beneath the ice. This introduces a significant uncertainty, particularly in warmer waters, where the impact on SIP estimates is greater. HB methods can be further categorized into those that rely on thin ice thickness and those that use sea-ice concentration (SIC). An example of the latter is Macdonald et al. (2023), who also employed AMSR-2 SIC data at a 3.125 km resolution, similar to our study. This relatively high resolution is critical for resolving small-scale polynyas and is therefore essential for our analysis.
Compared to Macdonald et al. (2023), we use a slightly simplified equation to calculate heat loss, primarily by neglecting downward solar radiation. These simplifications—specifically the exclusion of ocean heat flux and downward solar radiation—likely result in an overestimation

of SIP in our study. However, this overestimation does not affect the temporal variability and has only a minor influence on the seasonality of the estimates.

Direct comparisons of our results with previous studies are not possible due to differences in the regions and time periods considered. However, we generated a circumpolar dataset spanning 1992–2023 using SSM/I SIC data at 12.5 km resolution (Kaleschke, 2024). These results are comparable to previously reported values and broadly align with earlier estimates derived from the same sensor family (Janout and Kaleschke, 2024).

We further compared SIP estimates derived from SIC data at 12.5 km and 3.125 km resolutions, as well as from different reanalysis forcings (ERA5 and JRA55). Our findings indicate that while the seasonality and variability of the estimates are consistent, their magnitudes differ by up to a factor of two.

Janout, M., & Kaleschke, L. (2024). Gridded European circumpolar sea ice production fluxes (D1.4). Zenodo. https://doi.org/10.5281/zenodo.14192263

Kaleschke, L. (2024). EU project OCEAN:ICE Deliverable: D1.4 Gridded European circumpolar sea ice production fluxes [Data set]. Zenodo. https://doi.org/10.5281/zenodo.11652686

In the Method section, we added the following comparison:
L. 172–176: *"This study employs a heat budget method similar to previous approaches (e.g. Ohshima et al., 2016; Tamura et al., 2016) but with differences in methodology and satellite data. While this method likely overestimates sea-ice production due to simplifications such as neglecting ocean heat flux and solar radiation, it effectively captures temporal variability and seasonality. Comparisons with other resolutions and reanalysis forcing reveal consistent patterns in variability but differences in magnitude by up to a factor of two."*

[Figure]

Fig.: Monthly sea ice production for selected polynyas from two data sources: (1) SSM/I 12.5 km ASI ice concentration with ERA5 wind and air temperature (blue), and (2) AMSR-2 3.125 km ASI ice concentration with JRA55 wind and air temperature. The solid line shows the 2012–2023 average, and error bars represent interannual standard deviation.

L. 216–217: "It is easy to imagine that the varying intensity of the tides modulate the efficiency of the sea ice formation in coastal polynyas, as well as the exchange of water masses on the continental shelf with the ice shelf cavity, explaining the characteristic time scale of the observed variability."

Reviewer:
- This is an intriguing possibility, but I would expect tidal lateral excursions to be small relative to the size of the coastal polynya. Perhaps the authors could provide a more quantitative estimate to support their speculation?

Authors' response:
In the revised version, we removed this sentence and provided a quantitative estimate of the tidal excursion range and cavity overturning time scale:

L. 198–203: *"The PALAOA CTD is located approximately 40 km northeast of the ApRES location. Tidally modulated flow velocities of 0.4–0.5 m/s were inferred at this site (Ivanciu, 2014), with evidence of intense mixing below the northeastern part of the ice shelf (Smith et al., 2020). Tidal model simulations (Padman et al., 2008) that align with observations at the Ekström Ice Shelf (Fromm et al., 2023) predict spring tide velocities of 0.2–0.3 m/s in the vicinity of the ApRES location. Given that the tidal energy is concentrated at the diurnal and semi-diurnal frequencies, this indicates tidal excursions in the order of 10 km/d, suggesting that water masses from the PALAOA site may influence the ApRES location at time scales of days."*

Fig. 4

Reviewer:
- What is the time scale for the establishment of the plume circulation? Based on the ice shelf length and the plume velocity I would estimate O(10 days), which is much shorter than a seasonal time scale. Thus it is reasonable for seasonal variations in sea ice production (but perhaps not weekly?) to influence melt rates via changes in the plume velocity.

Authors' response:
Although not explicitly included in the steady-state model solution, the equilibration time scale corresponds to the advection time scale of the cavity overturning and it is indeed much shorter than the seasonal time scale (see response above).

- Also, the authors discuss the density contrast between the plume and the ambient water, but do not show profiles of the plume density. I suggest including these in the plot to aid the discussion.

Authors' response:
The plume density corresponds to the ISW density that can be derived from a given source water mass inside the cavity. The potential range of ISW densities originating from the expected inflow water masses is now included in the discussion of the T-S diagram of the cavity water masses.

- Have the authors considered running simulations with spring-like and autumn-like thermohaline stratification? It is surprising that spring does not receive particular focus here, despite exhibiting the highest melt rates. Is there insufficient hydrographic data to construct a full seasonal cycle of ambient stratification profiles in the cavity?

Authors' response:

Thank you for raising this point. The distinction between summer and winter was not well described here. What we wanted to represented with the simulation are the seasonal extremes in terms of ocean density in the cavity. Since these occur at the end of the polar summer and the end of the polar winter, it is more precise to assign the simulations to spring and autumn, rather than (polar) summer and (polar) winter.

There exists only one snapshot of hydrographic profiles from Jan 2019 below EIS. To mediate this shortcoming, we now include the seasonal water mass variations from the PALAOA observatory, which confirms the seasonal extrema that were used in the previous version of the manuscript. Those corresponded exactly to spring and autumn values, as is being requested by the reviewer.

In the revised version, we show the seasonal cycle of temperature, salinity and density based on the PALAOA dataset recorded between 2006 and 2014 near the ice shelf front. The seasonal cycle shows that the density minimum occurs in autumn and the maximum in spring. We used these seasonal extremes in the simulation.

Section 4.3: Plume model

Reviewer:

- How sensitive are the plume model results to the particular geometry or particular stratification used as inputs? The authors have selected a particular track along the ice shelf, and have made a specific choice in idealizing the summer/winter stratification, and I think it is important to assess whether these (somewhat arbitrary) choices have influenced the results. I would not expect a strong sensitivity to either choice, but I think the authors should check and discuss this.

Authors' response:

The reviewer is correct in assuming that choices of geometry and details of geometry have little influence on the overall results of the model simulations. We have added these explanations when discussing the T-S diagram (choice of ambient water mass profiles in the model) and when discussing the limitations of the plume model (weak sensitivity to geometry choice). To be fair, the chosen geometry corresponds to the only transect that has been reliably mapped for this ice shelf.

In the revised version, we wrote added a few sentences about the limitations of the plume simulation:

L. 393–394: *"Running the plume model with various (idealised) geometries confirms the robustness of results about the influence of the cavity stratification in the simulated melt rates regardless of choice of the detailed flow path of the plume."*

L. 228–234: "In summer, the low-density contrast between the rising ISW and the AASW that is pushed below the ice shelf weakens the meltwater plume by reducing its velocity, or even lead to the detachment of the plume from the ice base. Because the meltwater plume velocity is reduced and the additional heat provided by AASW is small at the ApRES site, melt rates are low in summer. In winter, sea-ice formation in front of the Ekström Ice Shelf erodes the stratification on the continental shelf and replaces the AASW inside the cavity with a homogeneous body of ESW. The higher density contrast between the ambient ESW and the rising plume is sufficient to drive turbulence at the ice–ocean interface. This increases the heat flux into the ice and thus enhances basal melting. When the plume rises further, the pressure decreases and the temperature of the plume can fall below the freezing point."

Reviewer:
- How large are the density contrasts between the plume and ambient waters in the winter and summer cases?

Authors' response:
In the revised version, this has been addressed by the discussion of the T-S diagram, which shows that the density differences due to seasonally varying upper ocean water masses are several times larger than the density difference that can be introduced by the pressure-dependent ISW formation.

- Also, this is an appealing dynamical explanation, but it is difficult for a reader to see exactly how the changes in ambient stratification influence the plume velocity without appropriate equations that express the plume dynamics. I recommend including at least the plume momentum equation to help clarify this mechanism. A diagnosis of the terms in the plume momentum balance in the summer and winter cases would provide quantitative backing for this conclusion.

Authors' response:
We believe that the formulation of the Jenkins (1991) ice shelf plume model has been established textbook material, and with little insight gained from restating the original dynamic equations. Instead, we augmented the model description with a conceptual explanation of the dynamics and refer the reader to the original paper for the detailed formulation. The dynamics are entirely controlled by the density difference between the plume and the ambient ocean properties, which are now being addressed together with a discussion of the T-S diagram. Conceptually, the plume gains buoyancy, because the meltwater input decreases its density compared to the salinity stratified ambient ocean, while melt rates are parameterized as a function of the plume velocity and the difference between the plume temperature and local melting point based on a shear-driven turbulent heat flux formulation.

**Authors point-to-point response on Referee Comment #2 to egusphere-2024-2109**

Zeising et al. present a new, almost 4-years long time series of basal melt rates from a site at Ekstrom Ice Shelf. The melt rates are inferred from an in-situ, phase-sensitive radar. It is pointed out that melting is higher in the winter months than in the summer months. The provided explanation of this seasonal difference is based on an assumed seasonal stratification change on the continental shelf. A meltwater plume model forced by two idealized profiles is used to support this claim. The plume model output is used to attribute melt rate change to either temperature or flow speed change.

The melt rate data the authors acquired and present are unique and very intriguing and I am excited by their appearance. On the oceanographic interpretation side of things, I think there are quite a few gaps and perhaps misunderstandings, furthermore, the most interesting aspects of the data don't seem to be addressed at all. I think with some more careful attention to the oceanographic interpretation this paper will be a nice contribution to the literature.

Some comments are below, others are included in the attached pdf.

* The focus of the paper is on seasonal melt rate variability, which the authors grossly simplify to higher melt in the winter and lower melt in the summer. The interpretation provided would be appropriate if the authors only possessed two melt rate measurements, one from each season and nothing else. However, the more complete data set, requires more elaborate explanations.

Some questions that aren't addressed:

1) What are the approximately monthly oscillations in the melt rates?

Authors' response:
In the previous version, we mentioned the occurrence of melt events in winter and spring, that cause the highest melt variation. In the revised version, we additionally mention now the monthly variability:

L. 247–245: *"A variability on monthly time scales is most pronounced when the largest absolute melt rates occur in winter and spring. During this time of the year, the difference between the minimum and maximum melt rate within a 28-d period exceeds 1 m/a.
In summer and autumn, the variability within the same time window is less than 0.3 m/a. The largest melt events that are observed from July/August until October/November, occurred with intervals ranging from two to four weeks, with an average duration of 21 d. At the beginning of these melt events, the melt rate increased abruptly to 1 m/a and more."*

2) In the winter, there are times when melt rate is as low as in the summer - how do you explain that, if the winter stratification is low - presumably AASW layer is formed in the summer only?

AND

3) While the monthly oscillations are also present in the sea ice production estimates, they do not seem to correlate well with the melt rate time series - why could that be?

Authors' response:
Based on our hypothesis, an influx of denser water causes an acceleration of the plume velocity, resulting in enhanced melt rates. The denser water erodes the stratification formed during the summer and autumn due to the inflow of the AASW. We hypothesize that the observed melt events are a consequence of transient inflows of dense water from sea-ice formation that were insufficient to erode the stratification, and cause a greater density contrast for a limited period. Consequently, the melt rates return to the summer level as long as the stratification remains intact. We expect that pulses of dense water that propagate through the cavity also interact with the time varying lateral circulation, such that the ApRES location is periodically exposed to more or less stratified environment, e.g. when spring tides mix dense water from the shallow bank in the east toward the central part of the cavity.

We added the following sentence to the Discussion:

L. 370–373: *"At the beginning of Winter, before dense water inflows have eroded the stratification within the entire cavity, transient fluctuations that enhance the density contrast for a finite duration, are causing the observed melt events."*

4) Although melt rate and sea ice production both show seasonal signal, there does not seem to be any interannual or intraseasonal correlation there. There are a lot of seasonal signals in the system, and it is not clear to me, or it hasn't been discussed why the sea ice signal is the main driver of the seasonal melting. Also, shouldn't there be a time delay in seasonal melt increase after the seasonal sea ice production start corresponding to the dense water propagation to the grounding line?

Authors' response:
Many thanks for these questions. In the revised version, we extended the explanation and description of the hypothesis. For instance, we addressed both in the discussion:

L. 380–384: *"In the hypothesis presented, the formation of dense water through sea-ice formation emerges as a driver for the erosion of stratification. The direct comparison between sea-ice growth and melt rate revealed moderate correlations. Although a consistent temporal lag was not evident, an estimated advection time scale of 8 to 16 d from the ice front to the grounding line may well explain the observed approximate phase shift between the sea-ice formation and the melt rate response, as the dense water signal propagates through the cavity."*

\* In general the argumentation should be a bit more specific, detailed, and supported. The following sentence, one of the few addressing the monthly oscillations I refer to above, is a good example of the vagueness of the explanations:

'It is easy to imagine that the varying intensity of the tides modulate the efficiency of the sea ice formation in coastal polynyas, as well as the exchange of water masses on the continental shelf with the ice shelf cavity, explaining the characteristic time scale of the observed variability.'

Authors' response:
We agree that some sentences were too vague. We have improve this in the revised version. The sentence in question has been removed.

\* The plume model gives a steady state solution. The authors force it with T/S profiles at the calving front (if I understand well). It will take a while for a change in continental shelf properties to propagate into the cavity - what is the time scale of that, and do the monthly oscillations interfere with that timescale, and therefore the applicability of the plume model to explain temporal variation?

Authors' response:
The CATS tidal model that predicts tidal current strength of 20-30 cm/s at spring tide period in the region of the drilling site, whereas tidally modulated flow speeds of 30-50 cm/s were inferred at the Palaoa site (Ivanciu, I. (2014)). With tidal energy being concentrated at the diurnal and semi-diurnal frequencies, this indicates tidal excursions in the order of 10 km/day, suggesting that water masses from the 30 km distant ice front may reach the ApRES site at time scales within days. However, the diffusive nature of such tidal transports is expected to become less relevant on horizontal scales that exceed a few tidal excursions, where the coherent background flow becomes more important. With typical flow speeds of 10-20 cm/s, the advection time scale between the ice front and the approximately 140 km distant grounding line is in the order of 8 to 16 days, which may well explain the phase shift between the sea ice formation and the melt rate response, as the dense water signal propagates through the cavity.

Ivanciu, I. (2014): On the role of supercooled water from beneath the Ekström Ice Shelf in the formation of a sub-ice platelet layer in Atka Bay, Antarctica, Bachelor thesis, Jacobs University Bremen, Alfred-Wegener-Institut Helmholtz-Zentrum für Polar- und Meeresforschung.

\* Some more justification of the realism of the idealized TS profiles would be useful. It looks to me that the summer profiles from Smith 2020 actually look more like your winter profiles - why is that?

Authors' response:
Thank you for raising this point. The definition and application of the summer and winter simulations have not been adequately described here. The simulation we present represents the seasonal extremes in ocean density within the cavity. These extremes

occur at the end of the polar summer and the end of polar winter, making it more accurate to assign the simulations to spring and autumn rather than (polar) summer and (polar) winter. In the revised version, we present the seasonal cycle of temperature, salinity, and density based on the PALAOA dataset recorded between 2006 and 2013 near the ice shelf front. The seasonal cycle demonstrates that the density minimum occurs in autumn and the maximum in spring. These seasonal extremes were utilized in the simulation.

As input to the model, we employed T/S profiles derived from observations at the site of the ApRES measurement. These CTD measurements were conducted in early summer 2019. However, at that time, the cavity at the ApRES site was still filled with cold and saline water, corresponding to the winter/autumn scenario in our simulation. Later in the year, salinity is anticipated to decline, and temperature to experience a slight increase. Based on PALAOA data from the calving front, the ocean density starts to decrease in November and December but reaches its minimum in late summer and fall.

To investigate the mechanism underlying the seasonality of the melt rate, we employed the plume model. However, to investigate short-term melt events that persist for only a few weeks, a more sophisticated model would be necessary which goes beyond the possibilities of this study.

Also, since the results will be sensitive to stratification, can you consider a range of plausible stratification profiles (also non linear ones) to capture some uncertainty in that, and therefore some sense of how robust the results are? Presumably AASW will not necessarily produce a linear profile if it restratifies the water column primarily near surface?

Authors' response:
We have considered both non-linear profiles (which primarily affect the upper part of the ice shelf, as properties in the lower part of the cavity remain unchanged) and less extreme seasonal variations (which give more moderate melt rate changes). In the revised version of the manuscript, we explicitly report on the case where ambient water masses resemble the observed ice shelf cavity profile and PALAOA conditions above.

* If the point of the paper is to distinguish between the effect of temperature vs speed on melt rate change, then that should be quantified somewhere.

Authors' response:
We agree that a discussion of the quantification of the influence of temperature and plume velocity on the melt rate is helpful. We added a figure and its detailed description in the revised version:

L. 307–323: *"To further evaluate the impact of changes in plume buoyancy, we analyse the influence of ambient temperature and ocean velocity on the modulation of the melt rate, based on the three-equation formulation (Hellmer and Olbers, 1989; Jenkins and Doake, 1991) that is used for parameterising heat fluxes at the ice ocean interface in the plume model. For instance, to achieve a melt rate of 0.24 m/a that is observed during summer, a*

*plume velocity of 0.015 m/s is required when assuming a water temperature of -2 °C (Fig. 5), as observed at the ice base during the early summer of 2018/19 (Smith et al., 2020). Higher melt rates, as are observed during late winter and spring, can be attributed to an increase in plume temperature and/or flow velocity. To obtain the average melt rate in spring of 0.71 m/a, a 0.2 °C higher plume temperature would be necessary to maintain a constant plume velocity. Conversely, an increase in plume velocity to 0.04 m/s with a constant plume temperature of -2 °C can also achieve the same melt rate. Both the plume temperature of -1.8 °C and the velocities of 0.04 m/a appear to be within a plausible range. However, for a maximum observed melt rate of 1.81 m/a, a higher temperature of up to -1.3 °C would be required to maintain a constant plume velocity of 0.015 m/s. Such a high plume temperature is not anticipated at the Ekström Ice Shelf, which has the highest weekly median temperatures in the PALAOA CTD around -1.7 °C, during a period of the year when the observed melt rates are lowest. If we assume that an increase in temperature alone was not the sole cause of the high melt rate, then the flow velocity of the plume must have also increased. At a constant temperature of -2 °C, an increase in the flow velocity to 0.11 m/s could have also resulted in a melt rate of 1.81 m/a, which still falls within a plausible range. This analysis suggests that the increase in melt rates in winter and spring cannot be attributed solely by warmer water, but rather to higher plume velocities."*

[Figure]

Fig.: Basal melt rate as a function of plume temperature and velocity, calculated using the three-equation model of ice-shelf ocean interaction (Hellmer and Olbers, 1989; Jenkins and Doake, 1991). The solid lines represent the contours for the average melt rate in summer (0.24 m/a) and spring (0.71 m/a), as well as the maximum observed melt rate of 1.81 m/a.

The horizontal dashed lines denote the grid for plume temperatures of −1.8 and −2 °C, while the vertical lines indicate the plume velocities corresponding to the indicated melt rates.

* It feels in the discussion as if the authors are trying to say their observed case doesn't fit standard understanding of sub ice shelf circulation. They say that what they observe is not Mode 1 circulation and they argument with variability and importance of flow speed changes (around line 238). Mode 1 circulation is a conceptualization of circulation that fits Ekstrom well, from the information provided. Mode 1 conceptualization does not distinguish relative importance of temperature vs speed changes on melt rate change. The idea of melt rate decrease as a consequence of circulation shut down due low sea ice formation has also been suggested before (Nicholls 1997), however was subsequently shown to be too simplistic (e.g. Hellmer 2012 and Naughten 2021). So it is probably worth highlighting similar concerns somewhere in this paper and not be overly conclusive.

AND

L. 238: "Thus, this process differs from mode 1 described in Jacobs et al. (1992)."

Reviewer:

I think there is a bit of a misunderstanding on the following lines. This is still mode 1 melting what the authors describe. The conceptualization of Mode melting doesn't address variability in melt rates and therefore doesn't attribute the meltrate variability to either temperature or velocity changes.

Authors' response:
The general notion is that mode-1 melt rate variability is associated with the changing amounts of heat that are provided at the grounding line by more or less HSSW formation (traditionally described in front of FRIS or RIS of the Amery Ice Shelf). When HSSW production is low, these cavities tend to fill up with ISW, which near their deep grounding lines, is much colder than the HSSW, such that an increased HSSW inflow enhances the thermal driving (and hence melt rate increase) in the deeper part of the cavity.
Our results emphasize an aspect of the mode-1 melting, where not the varying provision of heat at the grounding line modulates the melt rates, but the enhancement of the circulation strength as a response to the varying stratification within the cavity. This is a slightly different process than the FRIS case, where melt rates increase when HSSW inflows replace the colder ISW near the grounding line. At the Ekström Ice Shelf, ESW and HSSW have temperatures at the surface freezing point, whereas changes in thermal driving due to their different salinities are negligible.
In fact, running the plume model with a profile where the lower cavity is filled with ISW at temperatures close to the in-situ freezing point, rather than the Winter Water profile, has negligible effect on the melt variability at the ApRES site. However, the particular statement was removed when rewriting the discussion.

\* Comparison and interpretation of similar measurements at Nivlisen are a bit odd/wrong. There is no shown basis for why the plume at Nivlisen should be less strong than at Ekstrom.

AND

L. 261–262: "The absence of seasonality further away from the ice shelf front in conjunction with the low melt rate of <1 m/a at Nivl Ice Shelf is consistent with less pronounced plume-driven melting throughout the year."

Reviewer:

This is not true. If anything it is just evidence of lower seasonal variability of the plume dynamics, but not of a weaker circulation all together.

Authors' response:
Thanks for raising this point. We agree that it is not correct to say that the ApRES measurements at Nivl Ice Shelf indicate a less pronounced plume-driven melting as it rather indicates less seasonal variation. In the revised version, we removed the sentence.

\* Some indication of horizontal circulation beneath Ekström would be useful, since the authors use related platelet observations to support hypothetical ISW outflows. Related to that, some arguments why a simple 1D plume model is appropriate to model the 2D melt rates would be useful too. Are there perhaps any reliable mean satellite melt rate estimates that would support an across shelf uniform melt rate pattern?

Authors' response:
Thanks for raising this point. In the revised version, we added the following to the discussion:
L. 387–394: *"In reality, the one-dimensional overturning captured by this model will be affected by rotation at the cavity scale, where the ISW flow would be deflected to the left to follow geostrophic contours and rise where it is constrained to flow to upslope topography and Feltham, 2006). A more sophisticated model of the Fimbul Ice Shelf cavity (Hattermann et al., 2014) shows that such constraints may be provided by a central keel that constrains an upward flow of ISW along its eastern flank. A similar keel-like topographic feature is observed along the central flowline of the Ekström Ice Shelf (Fig. 1b), and with the ApRES being located along its eastern flank. However, running the plume model with various (idealised) geometries confirms the robustness of results about the influence of the cavity stratification in the simulated melt rates regardless of choice of the detailed flow path of the plume."*

Comments in the pdf:

L. 11: "Upscaling these observations to other ice shelves in this Antarctic sector will improve the overall assessment of the ice-shelf mass balance and **improve future projections**."

Reviewer:
- That would be great if it worked out that way, but so far there isn't a good evidence for that and definitely not something addressed in this paper

Authors' response:
We removed the sentence in the revised version. However, we also want to point out that in the framework of the NECKLACE initiative and other international observational programs the goal is to obtain a continental network of ice-shelf basal melt time series based on ApRES and use those to improve the model representations.

L. 45: "Instead, Eastern Shelf Water (ESW) with temperatures **close to the surface freezing point** is **assumed** to be the dominant water mass that interacts with the DML ice shelves (Vernet et al., 2019) [...]."

Reviewer:
- HSSW also has temperature close to the surface freezing point - the main difference between HSSW and ESW is in salinity.
- Assumed based on what? Is that a modeling result? Are there any observations available?

Authors' response:
In this context, the difference between HSSW and ESW is that HSSW is denser than the WDW (or CDW in other regions), by that forming the densest water mass in the region. In case of the ESW, any WDW that may enter the continental shelf would flow below the ESW towards the grounding line, initiating a mode-2 type of melting.
We referred to a review article of Vernet et al. 2019, which summarises this aspect of the eastern Weddell Gyre in an own subsection, but there are numerous individual studies that confirm this statement based on observational evidence, model results and conceptual reasoning, many of them being cited throughout this article (as well as in the cited review article). We have clarified these misunderstandings by rephrasing the sentence.

L. 48–50: "Basal melt rates from satellite remote sensing revealed in general low melt rates of 0.8±0.3 m/a on average **but higher melt rates exceeding 15 m/a at the deep grounding lines** of the Jelbart, Fimbul and Roi Baudouin ice shelves [...]."

Reviewer:
- Can you put it in the context of previous paragraph and state which mode of melting this indicates?
- How trustworthy are the satellite remote sensed estimates near grounding line?

Authors' response:
The higher melt rates near the grounding line can not be linked to a mode of melting without a further analysis of the temporal variability, which is not reliably possible based on the existing satellite-based melt rate products. Satellite based melt rate products can not be trusted close to the grounding line where the ice is not freely floating and lateral (non-local) strains have a major influence. However, it is not trivial to estimate the distance from the grounding line from where the ice is floating freely. Therefore, we cannot exclude that the high melt rates exceeding 15 m/a were estimated inside or outside this area. By reducing the mentioned limit to 5 m/a, we refer to a larger area at a greater distance from the grounding line, where the ice is at least partially freely-floating.

L. 64–66: "Sun et al. (2019) observed the basal melt rate near the grounding line of Roi Baudouin Ice Shelf and found nearly no melting in winter and highest melt rates of up to 10 m/a in summer which they linked to topographic ocean waves based on oceanic observations."

Reviewer:
- What mode of circulation did Roi Baudouin show based on their observations?

Authors' response:
Thanks for your question. The mechanism found by Sun et al. (2019) neither contradicts or supports any of the Jacobs-modes, as the waves can happily coexist with either of them.

We have included this in the revised version as follows:
L. 67–73: *"Sun et al. (2019) observed the basal melt rate near the grounding line of Roi Baudouin Ice Shelf and found nearly no melting in winter and highest melt rates of up to 10 m/a in summer. No oceanic measurements were available to determine whether WDW could reach this location, but the moderate melt rates indicate primarily a mode-1 driven melting near the grounding line, while the analysis of Sun et al. (2019) hypothesised that the melt rate variability was caused by seasonally enhanced propagation of tidal oscillations into the ice shelf cavity, which increase the turbulent heat transfer near the ice base.*

L. 66–69: "**Accordingly, the three ice shelves all show increased melt rates in summer**, partly on the **ice shelf front** (Fimbul and Nivl), but also near the grounding line (Roi Baudouin), albeit due to different processes."

Reviewer:
- Just two lines above you wrote that at Nivl, 35 km from the front, the highest melt rate occurred in winter
- At the front a summer peak is expected if it is mode 3, so not surprising

Authors' response:
It is true that one of the two ApRES sites at Nivl Ice Shelf showed the highest melt rates in winter (despite no pronounced seasonality). However, the other ApRES was located just 4 km from the ice shelf front, which showed the highest melt rates in summer. We can just cite and refer to those observations but will not try to explain these.

L. 74: "Study Area"

Reviewer:
- This section still feels like an introduction - it doesn't actually provide any info about the surveyed points and locations. I think it would be better if it was integrated within the introduction.

Authors' response:
We moved the text to the introduction.

L. 80–81: "The **minimum seafloor depth of 320 m** is located on the continental shelf break [...]"

Reviewer:
- Minimum depth is the most shallow point - is that what you mean? Or did you mean the deepest point - which would better fit into the WSW access discussion.

Authors' response:
Yes, we meant the "shallowest point". We corrected this in the revised version as follows:

L. 80–83: *"The most shallowest point of the seafloor with a depth of 320 m is located on the continental shelf break which is sufficiently shallow to prevent the inflow of WDW that circulates along the shelf break into the cavity below Ekström Ice Shelf (Eisermann et al., 2020)."*

L. 84–85: "Observations from sub-ice-shelf conductivity temperature depth (CTD) profiles showed that **most of the cavity** below Ekström Ice Shelf is filled with relatively cold ESW [...]"

Reviewer:
- The profiles were taken near the ice front and on the side close to the station - so the term most of the cavity is deceiving.

Authors' response:
We agreed and added the location of the measurement in the sentence:

L. 84–87: *"Observations from sub-ice-shelf conductivity temperature depth (CTD) profiles taken 30 km from the ice front showed that the cavity below Ekström Ice Shelf is filled with relatively cold ESW (in situ temperatures of −1.9°C, practical salinity of 34.4), while a buoyant plume of ISW (colder, less saline) is present near the ice-shelf base (Smith et al., 2020)."*

L. 86–88: "The Warm Deep Water that is suppressed below the continental shelf break in this region, is not expected to enter below the ice shelf (Smith et al., 2020)."

Reviewer:
- It wasn't observed - but why is it not expected to even intrude if it does elsewhere along DML ice shelf cavities?

Authors' response:
The bathymetry beneath the Ekström Ice Shelf does not feature any topographic depressions through which the WDW might enter the cavity as it has been observed to do elsewhere along the DML ice shelf cavities. We adjusted the sentence to clarify this:

L. 87–90: *"The WDW potentially provides ocean heat for ice shelf melting. However, the WDW is suppressed by the Antarctic Slope Front to depths around 500 m (Hattermann, 2018), which is generally deeper than the continental shelf break in this region. There is evidence that warmer inflows are restricted to topographic depressions (Lauber et al., 2024), which do not exist at Ekström Ice Shelf (Smith et al., 2020)."*

L. 88–90: "However, observations in front of the ice shelf (Nøst et al., 2011) and below the neighbouring Fimbulisen Ice Shelf (Hattermann et al., 2012; Lauber et al., 2024) indicate that remnants of fresher, solar-heated AASW may enter several hundred meters below the ice shelves along the DML coast."

Reviewer:
- 'However' feels like you'll keep talking about WDW but than you talk about AASW

Authors' response:
We agreed and removed "However".

Fig. 1:

Reviewer:
- I think it would be useful to add locations of CTD stations taken in the past (Smith 2020) to this map, at least to give the reader an idea how distant in space are the melt rate and oceanographic measurements.

Authors' response:
Thanks for this suggestion. We added the CTD locations to the figure. The ApRES was located at one of the CTD sites, but the ApRES time series started more than two years after the CTD measurements.

L. 121: "Since the ApRES was located below the surface, the measured ice thickness was not affected by snow accumulation."

Reviewer:
- Even if it was located at the surface, it still should not be affected by snow accumulation right? It would only snow on top of it anyway.

Authors' response:
That's correct. We will remove the sentence.

L. 131–133: "For the internal segments, we calculated the complex cross-correlation of the first measurement ($t_1$) **with the same segment of each repeated measurement ($t_i$)**. We used the lag of the highest amplitude correlation coefficient to find the correct phase-shift minimum, the sum of which gives the **vertical displacement**."

Reviewer:
- Not sure what you mean here. if you fix t1 then the same segment will only give you 1 pair.
- Displacement of what?

Authors' response:
We agree that these sentences were not well formulated. We fix t1 and did a cross-correlation of this depth-segment with every other measurement. In this way we obtain a vertical displacement time series of every depth-segment.

In the revised version, we will write this sentence as follows:
L. 133–135: *"For each internal segment, we obtained the displacement time-series from a cross-correlation of the complex signal of the first measurement ($t_1$) with each repeated measurement ($t_i$). We used the lag of the correlation coefficient with the highest amplitude to find the correct phase-shift minimum, the sum of which gives the vertical displacement of this segment since the time of the first measurement."*

Eq. 5:

Reviewer:
- Did you assume constant strain rate through time or did you derive strain rate timeseries? If the letter, could you show those somewhere in the paper?

Authors' response:
We derived a time series of the cumulative thinning due to strain and firn compaction. We subtracted both from the cumulative change in ice thickness to obtain the cumulative melt. Thus, we did not derive a strain rate time series. We added a figure showing the cumulative time series. It shows that the strain is nearly constant over time, roughly about 0.0024 1/a.

[Figure]

Fig.: Cumulative change in ice thickness (ΔH), attributed to strain and firn compaction (Δ H_f + Δ H_ε) as well as basal melting (Δ H_b) from April 2020 to November 2023. The vertical stripes denote the respective seasons.

L. 149: "[...] the ApRES was locally repositioned, a few metres laterally away from the previous location [...]"

Reviewer:
- Just the instrument or also the antennas? - did you use new antennas each time?

Authors' response:
We repositioned the ApRES and the antennas. We always used the same skeleton slot antennas as they are more robust than the more fragile bow-tie antennas. In the revised version, we added that we have also repositioned the antennas.

L. 150–152: "To determine the melt rate at a certain point in time, we smoothed the **data** by applying a 36 h **moving average filter**, and calculated the basal melt rates based on the gradient in a 7 d **moving window**."

Reviewer:
- What do you mean by data? displacement timeseries?
- Moving average is actually a very poor low pass filtering choice and doesn't give good control over the filtered bandwidth - consider using a different filter (e.g. butterworth)
- Where did you apply this filter? to the melt rate timeseries? to the displacement timeseries?

Authors' response:
Thanks for your questions. We agree that this part was not well formulated. Our analysis is based on a well established processing scheme for ApRES time-serieses, considering cumulative ice thickness change, firn compaction and strain thinning. We first subtracted the firn compaction and the strain thinning from ice thickness change to obtain the time-series of cumulative basal melt. We smoothed the cumulative time series of the basal melt with a 36h low-pass filter. Previously, we used a moving average filter that we now changed to a butterworth filter. However, there is no significant difference in the result. To obtain a time series of the melt rate, we used a 7d moving window to calculate the averaged melt rate from the gradient within this time-window.

In the revised version, we changed this part as follows:
L. 154–156: *"We obtained the time series of cumulative melt by subtracting the time-series of firn compaction and strain thinning from the cumulative ice thickness change. To determine the melt rate, we smoothed the cumulative melt time-series by applying a 36h butterworth filter, and calculated the gradient in a 7d moving window which gives the 7d average melt rate."*

L. 153: "[...] we also calculated the 7d standard deviation of the basal melt rates based on the gradient in a 1d moving window."

Reviewer:
- Not clear what you did, please rephrase.
- Are you applying a 7-d moving average filter to some timeseries - clarify what frequencies that actually retains, or use a filter with a better defined bandwidth retention.

Authors' response:
We agreed that the sentence was not well written. We wanted to represent the variability of the melt rate on sub-weekly time scales. Therefore, we calculated the melt rate again, but this time based on the gradient in a 1-day window instead of a 7-day window. The standard deviation of the 1-day melt rate within a 7-day moving window is shown in Fig. 2.

In the revised version, we have rephrase this sentence as follows:
L. 157–158: *"To represent the variability of the melt rate on sub-weekly time scales, we additionally calculated the melt rate from the gradient in a 1 d moving window and calculated the 7 d standard deviation."*

L. 175–176: "[...] we assume that the idealised model captures the basic features of the cavity circulation of this ice shelf, while tides superimpose to further modulate the melt rates."

Reviewer:
- What do you mean by this? that you don't capture tidal effects? which particular aspect of tidal effects? Can you elaborate?

Authors' response:
We agree that this was not well expressed. In the discussion section of the revised version, we elaborated more on this point:

L. 398–341: *"The plume model also does not explicitly capture tidal effects, such as the increase of entrainment of ambient water into the plume or turbulent mixing at the ice-ocean interface. However, since these effects superimpose linearly on the simulated processes, we assume that the proposed impact of seasonal stratification on the plume dynamics is a robust result, while the detailed modulation of these dynamics remains subject for further studies."*

L. 190: Results

Reviewer:
- I think you need to highlight somewhere in the text the frequency of the melt event which appears (to me by eye) to be about a month. This is quite a striking feature and at least some attention needs to be given to it.

Authors' response:
We performed a frequency analysis that revealed that the melt events occurred at irregular intervals. The melt events lasted between two and four weeks. Thus, we could not determine a specific frequency that could be attributed to the tides.

In the results section, we mentioned the periods of the melt rates:
L. 249–251: *"The largest melt events that are observed from July/August until October/November, occurred with intervals ranging from two to four weeks, with an average duration of 21 d."*

Table 1:

Reviewer:
- I am not sure separation into 4 season makes a whole lot of sense when the time series appear to have only 2 distinct regimes.

Authors' response:
We prefer showing the melt rate separated into 4 seasons instead of two (summer and not-summer) as it shows better the seasonality with the increasing melt rate over Autumn winter and spring before the melt rate decreases before summer.

L. 205–206: "A comparison of the melt rate and sea-ice growth time series with their average values for the individual seasons shows a **similar** pattern"

Reviewer:
- I think more than 'similar', 'seasonal' is more appropriate, apart from the seasonal signal, there isn't actually much correspondence between the time series interannually or intraseasonally

Authors' response:
We agreed and changed the sentence so that it does not include the comparison anymore.

L. 215–216: "Generally, the time scale of this variability **resembles the fortnightly spring-neap cycle**, on which tidal currents strengthen and weaken in Atka-bay (Smith et al., 2009)."

Reviewer:
- Maybe show a power spectrum to support this? Or some other spectral plot. To me it looks more like a monthly variation than a fortnightly one.

Authors' response:
We have removed the section that contained this sentence.

L. 216–218: "It is easy to imagine that the varying intensity of the tides modulate the efficiency of the sea ice formation in coastal polynyas, as well as the exchange of water masses on the continental shelf with the ice shelf cavity, explaining the characteristic time scale of the observed variability."

Reviewer:
- Please describe the specific plausible physical processes, and if possible, provide some estimate on how much difference can each process explain for a given quantity of interest (e.g., volume of dense water formation, distance of intrusion into cavity etc)

Authors' response:
The volume of dense water formation is difficult to assess based on the sparse data at hand. In the revised version, the effects on water column stability are assessed based on the T-S diagram and the comparison of the Sub-EIS-Obs CTD profiles with the seasonal water mass evolution at PALAOA. Advection time scales are being assessed in the methods and further discussed in relation to the phases.

L. 220: "The simulation results show no significant differences in the plume temperature, but differences in the plume velocity (Fig. 4)."

Reviewer:
- How did you assess significance? I think you should quantify these in terms of percent change of melt rate as a response to a change in each of the variables (T and U) separately. The temperature difference is actually significant at the ApRES location, except that it produces the opposite effect of the winter velocity change. Which further supports your claim of velocity being the melt rate increase driver. So just write that out clearly.

Authors' response:
Thanks for your question. It is true that a slight change in temperature can increase the heat flux and that the basal melt rate. To discuss the role of plume temperature and velocity, we added a figure to the manuscript showing the impact of both on the heat flux (Fig. 5).

L. 226–228: "In summer, the **low-density contrast between the rising ISW and the AASW** that is **pushed below the ice shelf** weakens the meltwater plume by reducing its velocity [...]"

Reviewer:
- Isn't it more so the increase in stratification that causes the detachment - plume reaching neutral buoyancy deeper in the water column, than the density contrast between AASW and ISW?

Authors' response:
Yes, this is correct. We revised this section:
L. 353–356: *"The absence of the dense water formation and a restratification of the water column during the summer and autumn reduces the buoyancy of the ascending ISW, causing the arrest of the buoyant plume as it encounters water masses of the same density at the ApRES site."*

- You didn't actually model that - the intrusion of AASW beneath the ice shelf in the mode 3 sense

Authors' response:
Yes, this process is implicitly included in the plume model formulation. When the ascending ISW becomes neutrally buoyant and detaches from the ice base, a new plume initiates with properties of the ambient water at that depth. In the case of the AASW profile, this leads to increased plume temperature below shallower ice, compared to the Winter Water case, but, as can be seen in the plume model results figure, this effect is less important than the loss of plume velocity for the melt rates.

L. 229–230: "Because the meltwater plume velocity is reduced and the **additional heat provided by AASW is small** at the ApRES site, melt rates are low in summer."

Reviewer:
- It is actually less, the difference is not necessarily small at the observed site

Authors' response:
This sentence has been removed.

Fig. 4

Reviewer:
- Can you add T and S profiles from Smith 2020 to illustrate how well the idealized profiles agree with those data?
- It looks to me that the summer profiles from Smith 2020 actually look more like your winter profiles - why is that?

Authors' response:
We have added T and S profiles from Smith et al. (2020) to the figure. It is true that they correspond to winter profiles as they were measured in early summer, while the lowest density is likely to be reached in autumn. See the comment above for more details.

L. 234–236: "The super-cooled ISW plume is deposited as platelet ice beneath the sea ice in front of the ice shelf. This agrees with the results of Arndt et al. (2020), who found seasonality in platelet ice formation at Atka Bay with the highest accumulation rates in winter and spring."

Reviewer:
- Wouldn't you expect the ISW outflow on the western side of the ice shelf due to Coriolis - while Atka Bay lies east of it. If not - can you provide some indication of a plausible 3D circulation beneath Ekstrom that would result in outflow in the east?

Authors' response:
We acknowledge the valid remark of the reviewer and in lack of better evidence, our answer can only remain speculative. Regarding the leftward deflection of ascending ISW due to Coriolis, one would generally expect the plume to rise where it is constrained to flow to upslope topography (Holland and Feltham, 2007). A more sophisticated model of the Fimbul Ice Shelf cavity (Hattermann et al. 2014) shows that such constraints may be provided e.g. by the central Jutulstraumen keel that constrains an upward flow of ISW along its eastern flank (e.g. Fig. 8c in Hattermann et al. 2014). A similar keel-like topographic feature is observed along the central flowline of the Ekström Ice Shelf (Fig. 1b), and with the ApRES being located along its eastern flank. While ISW originates at the central part of the seismic profile that is indicated in Fig. 1b, we assume that an ISW plume that is initiated eastward of that central keel will at least partially be guided along the eastern part of the ice shelf (e.g. as a geostrophic flow following the 500 m contour). Another aspect is that the barotropic component of the cavity circulation will to large degree be steered by the water column thickness, rather than the respective slope of the ice base or seafloor. While there exist only a single along-shelf transect of water column thickness from the grounding line, the cross-shelf profiles shown in Fig. 3c of Smith et al. (2020), clearly shows the trough-like bathymetry which is expected to steer the geostrophic flow in along-shelf direction following the flank of the trough. Regarding the water mass exchange with Atka Bay, the zone of shallow water column thickness around Neumayer station and toward Atka Bay is expected to be subject to intense tidal mixing (e.g. profile EIS 4 in Fig. 4 of Smith et al., 2020). While we expect that tides mediate water masses between Atka Bay and the deeper part of the cavity, where the ApRES is located, it is also possible that the observed platelet ice in Atka Bay is of more local origin, where tidally enhanced mixing of AASW below the ice shelf maintains an efficient ISW formation.

L. 237–238: "The interpretation of the simulation suggests that the melt rate time series shows a reduced melt rate in summer while the **reactivation** of the plume restores normality."

Reviewer:
- Isn't there always some plume? what do you mean by reactivation

Authors' response:
When the stratification within the cavity generates a low-density contrast between the rising ISW and the ambient water, the plume's speed can be substantially reduced, potentially leading to its detachment from the ice base. During winter and autumn, when dense water erodes the stratification, the plume accelerates and generates turbulence at the ice-ocean interface. With "reactivation," we refer to the acceleration and attachment of the plume to the ice base. To avoid confusion, we will rephrase the term to "enhancement of the plume."

L. 241–242: "**In the case of Ekström Ice Shelf, we assume that most of the cavity is filled** with water close to the **surface freezing point** (Smith et al., 2020), **such that the thermal effect of sea ice melt water is small** [...]"

Reviewer:
- As far as I follow, I think the only difference for your modeling if there was HSSW instead of ESW would be a constant offset in salinity, and therefore density, but that would not change anything since it is the density gradient that matters here.
- HSSW is also close to freezing point
- I don't understand what you mean here. Please elaborate.

Authors' response:
We agree that this sentence was not well formulated. In the revised version, we rewrote this as follows:
L. 44–50: *"Due to relatively cold, fresh, and hence buoyant continental shelf water masses, the formation of dense High Salinity Shelf Water (HSSW) is limited here (Thompson et al., 2018), which is the driver of mode-1 melting in other regions of Antarctica (Nicholls, 2018). Instead, the lighter Eastern Shelf Water (ESW) has been observed to be the predominant water mass that interacts with the DML ice shelves (Vernet et al., 2019). Similar to the HSSW, temperatures of the ESW are also close to the surface freezing point. However, in contrast to the HSSW, the ESW is less dense than the WDW that is observed to enter through topographic depressions at the continental shelf break (Lauber et al., 2023). It provides additional heat for melting, while the solar heated and relatively fresh AASW may intrude deep below the ice shelf (Hattermann et al., 2014; Lauber et al., 2024) and affect the cavity circulation on seasonal timescales."*

L. 245–247: "In this regime, the temperature of the ambient water is only secondary to the increase or decrease in melt rate, while this highlights the role of the sea-ice formation in the **reactivation** of the plume."

Reviewer:
- I think increase in flow speed would be more descriptive than reactivation. Here and elsewhere in the MS

Authors' response:
We agreed and changed the wording.

L. 247–249: "We draw the conclusion that a reduced sea-ice formation in front of the ice shelf, such as widely observed in Antarctica in 2023, as well as a freshening of upper ocean properties, could result in a reduced mean annual melt rate."

Reviewer:
- Similar conclusions based on simple models and arguments have been drawn before (e.g. Nicholls 1997 Nature), but have been subsequently shown to be too simplistic (e.g. Hellmer 2012 and Naughten 2021). So probably worth highlighting similar concerns somewhere in the paper and not be overly conclusive.

Authors' response:
We removed this section in the revised version.

L. 249–250: "Hence, the effect of sea-ice formation and ice shelf cavity stratification will need to be **taken into account together** with other processes like potential warming of the AASW [...]"

Reviewer:
- Taken into account when doing what? for what purpose. Elaborate here on what you are refering to.

Authors' response:
We removed this section in the revised version.

L. 258–261: "While a similar effect might dominate the meltrate seasonality closer to the calving front of Ekström Ice Shelf, its more confined geometry and deeper grounding line promotes the development of **coherent ISW plumes** which modulate basal melting when interacting with the seasonally varying cavity stratification."

Reviewer:
- What do you mean by more coherent and what is the basis for this statement? Are you claiming that the presence of a strong plume prevents AASW intrusion beneath the ice shelf front? I don't think that is true. In fact it could even drive it in by enhancing circulation.

Authors' response:

In the revised version, we revised this section. There is no claim about the prevention of AASW intrusions, although the reviewers scenario where a strong plume may "drive in" AASW (in the upper part of the water column). It appears more reasonable that a strong plume is associated with an enhanced inflow of (denser than AASW) waters at depth.

L. 277: "Simulations with a meltwater plume model suggests that the seasonal **inflow** of Antarctica surface water enhances the stratification inside the cavity and weakens the plume."

Reviewer:
- You didn't simulate inflow, but change in stratification on the continental shelf

Authors' response:
That is true. We changes the sentence as follows:
*"Simulations with a meltwater plume model suggest that in summer and autumn, the enhanced stratification inside the cavity weakens the plume."*

---

## Author Response (AR2)

Dear Reviewer, dear Editor,

We would like to thank you once again for your efforts to improve our manuscript! We have implemented the majority of the suggestions mentioned by the reviewers and responded to the most important comments in more detail in the attached point-to-point response.

Recently, a paper about ApRES-derived melt rates on the Fimbul Ice Shelf from Lindbäck et al. (2025) was published. Since we had previously included two similar studies in the introduction, we have now also included this study in the introduction and in Fig. 1.

Thank you for your time and expertise.

Kind regards,
Ole Zeising and co-authors

**Authors point-to-point response on Referee Comment #2 to egusphere-2024-2109**

*L. 155–156: "To determine the melt rate, we smoothed the cumulative melt time-series by applying a 36h butterworth filter, and calculated the gradient in a **7d moving window** which gives the 7d average melt rate."*

Reviewer: 7-d moving window doesn't actually provide a 7 day frequency cutoff, which is the more relevant quantity. Can you provide the frequency cutoff in the final analyzed melt rate time series? Also, it is unclear why you first apply 36 h low pass filter and then do moving average when both act as low pass filters. If you want 7 day cutoff why don't you just do 7 d butter worth right away?

Authors' response:
The melt rate is based on the average gradient within the 7-d moving window of the cumulative melt time series. The reviewer is right, that the 36h butter worth filter does not change much here, so we removed it and updated the melt rate values. Calculating the average gradient within a 7-d moving window attenuates periods shorter than ~16 days. We added this to the manuscript.

*L. 176–177: "While this method likely overestimates sea-ice production due to simplifications such as **neglecting ocean heat flux and solar radiation, it effectively captures temporal variability and seasonality.**"*

Reviewer: Do the neglect variables such as ocean heat flux not have temporal (seasonal) variability that could interfere with the results? Or is there some reason to assume these effects are small?

Authors' response:
It is common practice to neglect ocean heat flux for the heat balance method. This is because the information is very sparse or highly uncertain. In fact, this heat flux is likely negligible unless it is a sensitive heat polynya, such as the Maud Rise Polynya (see e.g. Lin et al. (2023),  https://doi.org/10.1029/2022GL101859). During the main phases of sea ice production, solar radiation is almost zero and is therefore neglected in the model of Pease (1987).

*L. 177–178: "Comparisons with other resolutions and reanalysis forcing reveal consistent patterns in variability but differences in magnitude by up to a factor of two."*

Reviewer: Do you have a figure to show that?

Authors' response:
Direct comparisons of our results with previous studies are not possible due to differences in the regions and time periods considered. However, we generated a circumpolar dataset spanning 1992–2023 using SSM/I SIC data at 12.5 km resolution (Kaleschke, 2024). These results are comparable to previously reported values and broadly align with earlier estimates derived from the same sensor family (Janout and Kaleschke, 2024).

We further compared SIP estimates derived from SIC data at 12.5 km and 3.125 km resolutions, as well as from different reanalysis forcings (ERA5 and JRA55). Our findings indicate that while the seasonality and variability of the estimates are consistent, their magnitudes differ by up to a factor of two.

Janout, M., & Kaleschke, L. (2024). Gridded European circumpolar sea ice production fluxes (D1.4). Zenodo. https://doi.org/10.5281/zenodo.14192263

Kaleschke, L. (2024). EU project OCEAN:ICE Deliverable: D1.4 Gridded European circumpolar sea ice production fluxes [Data set]. Zenodo. https://doi.org/10.5281/zenodo.11652686

[Figure]

Fig.: Monthly sea ice production for selected polynyas from two data sources: (1) SSM/I 12.5 km ASI ice concentration with ERA5 wind and air temperature (blue), and (2) AMSR-2 3.125 km ASI ice concentration with JRA55 wind and air temperature. The solid line shows the 2012–2023 average, and error bars represent interannual standard deviation.

*L. 217–219: "**Omitting Coriolis' effect**, the system is parameterised along a one-dimensional flowline, described by four ordinary differential equations with prognostic variables for the plume thickness D, speed U, temperature T, and salinity S with details of the formulation being given in Jenkins (1991)."*

Reviewer: That doesn't seem to be an appropriate assumption for an ice shelf as wide as Ekstrom - what is the ratio of ice shelf with to rossby radius of deformation here?

Authors' response:
We acknowledge that the reviewer requests a more nuanced discussion of the applicability of the simple one-dimensional plume model. We addressed this in the manuscript. Here, the sentence provides an accurate description of the model formulation, which we prefer to leave it unchanged.

*L. 222–223: "Given the confined configuration of the Ekström Ice Shelf **along a quasi-one-dimensional flowline geometry**, **we assume that the idealised model adequately captures the basic features of the cavity circulation beneath this ice shelf.**"*

Reviewer:
- This might be the case for the ice flow, but not necessarily for the ocean flow - See, for example, numerous ocean modeling simulation results beneath a relatively small Pine Island Ice Shelf and the complexity of circulation there.
- I don't think you can make that argument based on scaling parameters (you can try or maybe you already have and if you succeeded, please provide the quantitative information in the paper). I understand the desire to use a simple model, but I think you need to acknowledge here in methods already that there isn't actually any basis for why it should capture the basic features of the 3D cavity circulation. You can still go ahead and use the simple model, but with being honest that the only motivation is its simplicity, and not actually the fact that it is justified or appropriate.

Authors' response:
We agree with the reviewer that the cavity circulation below Ekström Ice Shelf is likely influenced by rotational effects. In fact, we did neither state nor intend to claim that our idealized model captures the 3D cavity circulation (which seems to be the core criticism here and in other comments). While rising ice shelf water plumes indeed seem to manifest as (rotationally affected) near-geostrophic, topographically steered currents (Holland & Feltham, 2007), many three dimensional GCM simulations of (idealized and more complicated) of ice shelf cavities with similar confined, elongated flowline geometries such as the the Ekström Ice Shelf, show a systematic cavity wide circulation that can be conceptually be described as a quasi-two-dimensional overturning circulation, which manifests in three dimensions through topographically confined near-geostrophic currents – a recent example given by the MISOMIP simulations proposed by Asay-Davis et al. 2016 (https://doi.org/10.5194/gmd-9-2471-2016). Despite complexities at smaller scales, this overarching circulation is primarily driven by buoyancy forces due to water mass transformation from the ice-ocean interactions that are represented in the idealized one-dimensional plume model. Hence, we politely disagree with the reviewer, finding appropriate justification for the use of this model for representing the basic feature of and making it useful for studying the Ekström Ice Shelf cavity circulation, which provides additional motivation for its use beyond its convenient simplicity. To address the reviewer's concern, we revised the manuscript to clearly state the limitations of the model in the methods in addition to the discussion part:

*"The Ekström Ice Shelf comprises a confined ice geometry along a quasi-one-dimensional flowline. The plume model approximates the averaged cavity circulation across the ice flow to provide an initial assessment of the dynamics of buoyant ISW that rises along the upward-sloping ice base. However, the three-dimensional circulation beneath the ice shelf is likely influenced by rotational effects. Consequently, the model results are only applicable to regions where the flow is constrained by topography (Jenkins, 1991). Further limitations of this approach will be addressed in the discussion."*

*L. 235–236: "This end member was **linearly interpolated** to the observed water masses at 150 m (**constant to the surface**) derived from the PALAOA CTD time series, representing a late winter/autumn (Sept/Oct) WW extreme and a late summer/spring (Feb/Mar) AASW extreme."*

Reviewer:
- Why is linear vertical profile appropriate? Is that what the other available CTD profile suggests?
- What was constant and where?

Authors' response: A linear profile is consistent with the simplicity of our modeling approach that was pointed out by the reviewer (we specified this in the revised version of the manuscript). Existing profile data from Ekström and other ice shelves are ambiguous. In fact, we tried various profiles, e.g. prescribing AASW with a mid-depth pycnocline, or extrapolating the profile above the PALAOA CTD based on observed open ocean end-member values, only finding minor impact on the magnitude of our results, while the qualitative picture remains unchanged.
Updated sentence:
*"This end member was linearly interpolated to the observed water masses at 150 m derived from the PALAOA CTD time series (and with temperature and salinity kept constant above to the surface), representing a simplified late winter/autumn (Sept/Oct) WW extreme and a late summer/spring (Feb/Mar) AASW extreme."*

*L. 251–252: "The strongest melt event occurred in September 2020, when the melt rate increased from 0.21 to 1.81 m/a (**7-day average**)."*

Reviewer: Do you mean over 7 days duration? Else I don't understand what you mean

Authors' response:
We refer to the melt rate change between two consecutive 7-day bin-averaged melt rate estimates. Thus, the melt rate represents the average melt rate within 7 days. Within the 7-day window, even higher melt rate occurred at shorter times.

*L. 257: "Figure 2: Time series of 7-day average basal melt rate from autumn 2020 to spring 2023 (blue line) with **sub-weekly variability represented by the standard deviation (shaded area).**"*

Reviewer: Why don't you just show the melt rate times series before the 7 day averaging? It would make it clearer how abrupt is the onset of the melt rate events vs how much of it is smoothed out.

Authors' response:
With the approach we used to analyse the ApRES time series, we obtain a cumulative melt time series, from which we calculate the gradient within a 7-day window to get the melt rate. The cumulative melt rate is not an ideal signal. The 1-day average melt rate is to a certain extent affected by noise. Thus, we prefer showing the 7-day average melt rate.

*L. 261–262: "When the sea-ice concentration reached a coverage of 90% in April and May, the growth rate initially declines, but increases again towards the end of winter."*

Reviewer: What is this dip in Jun/July caused by? lower winds or warmer temperatures? Perhaps including a plot of those two variables would be useful too, for completeness.

Authors' response:
The bi-modality of sea ice production is a common feature for other polynyas (see Fig. above). We suggest that this seasonality is related to the maximum ice concentration as shown in Fig. 3 of the manuscript, which insulates the ocean from the cold atmosphere and limits further growth. A further attribution of the underlying reason is difficult due to complex sea ice dynamics and interaction of the ocean and the atmosphere.

*L. 268–270: "A cross-correlation analysis between sea-ice formation and melt rate, calculated for each season, revealed predominantly moderate (0.5 — 0.75) or low (0.25 — 0.5) correlation values **for various lags of up to 26d**."*

Reviewer: Do the lags have either sign, or do you always have?

Authors' response:
The lags are only positive (an increase in sea ice formation is followed by an increase in melt rate), which is described in the previous sentence.

*L. 327–329: "However, a detailed comparison of the melt rates determined from ApRES measurements in this study with those estimated from satellite observations is difficult due to different observational periods."*

Reviewer: But can you still show the other satellite estimates, side by side, even if they are for different periods? It would be useful to have that anyway, to see whether there is some substantial change in character of the time series using different techniques or not. Finally, can your temperature time series may provide some indication of melt rate variability or of the stability of the conditions, which may be useful for tying the different time periods together?

Authors' response:
Previous comparisons of ApRES and satellite based estimated time-series have shown large differences due to the uncertainties in the satellite-based method (Vanková and Nicholls, 2022). Since non-validated time series based on satellite observations are not trustworthy so far, we do not want to compare the characteristics of the different time series observed at different times. Unfortunately, this means that we can't use our time series to validate the satellite-based estimates as they don't overlap.

*L. 345–348: "In contrast to Fimbulisen and the Nivl Ice Shelf, the Ekström Ice Shelf also exhibits a more confined geometry and a deeper grounding line, which together promote the formation of **coherent quasi-two-dimensional cavity overturning circulation that** is driven by the pressure-dependent formation of buoyant ISW."*

Reviewer: As mentioned above, justify using scaling arguments, or leave out

Authors' response:
Perhaps there is a misunderstanding, where the term "quasi-two-dimensional" did not intend to refer to the absence of rotational effects which the reviewer seems to refer to in their comment. The intention with this formulation was to describe the expectation of a relatively clean-cut estuarine-alike circulation, with (primarily) one inflow pathway along the sloping bottom and (primarily) one outflow pathway along the sloping ice base, which coherently spans the entire Ekström Ice Shelf cavity (primarily oriented along the ice flow path, but skewed by Coriolis). This is opposed to the situation at Fimbulisen, where observations (e.g. Nicholls et al. 2006) show the existence of multiple inflow pathways, and 3-D simulations (e.g. Hattermann et al. 2014) suggests a horizontal circulation pattern that is affected by a more complex bottom and ice shelf geometry with a highly asymmetric distribution of deep and shallow ice (and the later being also true for Nivilisen).  To avoid ambiguity, we have dropped the term "quasi-two-dimensional" in our explanation in the revised version of the manuscript:
*"In contrast to Fimbul and the Nivl Ice Shelf, the Ekström Ice Shelf also exhibits a more confined geometry and a deeper grounding line, which together promote the formation of a coherent cavity overturning circulation that is primarily driven by the pressure-dependent formation of buoyant ISW."*

*L. 393–394: "Running the plume model with various (idealised) geometries confirms the robustness of results about the influence of the cavity stratification in the simulated melt rates regardless of choice of the detailed flow path of the plume."*

Reviewer: I thought the keel was focusing outflow because of rotational effects. Running plume model without rotation cannot account for geometric-rotational effects in any way.

Authors' response:
The assumption here is that (non-resolved) rotational effects focus the outflow along a near-geostrophic, topographically steered flowpath, as e.g. described in Holland & Feltham 2006, https://doi.org/10.1175/JPO2970.1), or seen in the 3-dimensional simulations of the Fimbulisen cavity (Fig. 8 in Hattermann et al. 2014, https://doi.org/10.1016/j.ocemod.2014.07.004). By prescribing the geometry along such a near-geostrophic flowpath, the geometrical-rotational effects are implicitly approximated in the one-dimensional plume model – and extreme localization of this concept is e.g. provided by Lazeroms et al. 2018 (https://doi.org/10.5194/tc-12-49-2018). Surely, the representation is incomplete, but at this point, we do not expect that a more sophisticated representation of the cavity circulation would make much difference to our main finding, which primarily relates the circulation strength inside the cavity to the ISW buoyancy relative to ambient water that enters the cavity, which, together with observational insights and inductive reasoning lead us to formulate a novel hypothesis for explaining observed melt rate variability below Ekström

Ice Shelf. We assume that a similar result could for instance have been obtained with a idealized model of the cavity overturning (e.g. following the approach of Walker & Holland 2003 (https://doi.org/10.1016/j.ocemod.2007.01.001)), or the box model formulation of Olbers and Hellmer (doi: 10.1007/s10236-009-0252-z), which are equally simplistic but conceptually diametral representations the cavity circulation compared to Jenkin's plume model. Using a more sophisticated 2D-plume model, or a fully resolved 3-D GCM simulation might be considered superior for more quantitative analyses, but those would face other challenges of e.g. providing more detailed accurate forcing and boundary conditions, including a well constrained three dimensional geometry of the cavity, as well as accurate representation of mixing processes inside the cavity (Gwyther et al. 2020, https://doi.org/10.1016/j.ocemod.2020.101569). Discussing all these aspects in detail may arguably be out-of-scope of this paper, which centers around (as was pointed out in previous reviews) a novel observational dataset of basal melt rates.

*L. 399–401: "However, since these effects superimpose linearly on the simulated processes, we assume that the proposed impact of seasonal stratification on the plume dynamics is a robust result, while the detailed modulation of these dynamics remains subject for further studies."*

Reviewer: There are definitely nonlinar tidal effects and currents in the system

Authors' response:
Agreed! We removed the term "linearily." A study that suggests how linear may be of leading order in some cases is shown in Jourdain et al. (2019), (https://doi.org/10.1016/j.ocemod.2018.11.001). Hence, we propose that the detailed modulation of these dynamics remains subject for further studies.